# SPACE: Noise Contrastive Estimation Stabilizes Self-Play Fine-Tuning for Large Language Models

**Yibo Wang**[1,2,*], **Guangda Huzhang, Qing-Guo Chen**[3],
**Zhao Xu**[3], **Weihua Luo**[3], **Kaifu Zhang**[3], **Lijun Zhang**[1,4,2,†]

[1]National Key Laboratory for Novel Software Technology, Nanjing University, Nanjing, China
[2]School of Artificial Intelligence, Nanjing University, Nanjing, China
[3]Alibaba International Digital Commerce [4]Pazhou Laboratory (Huangpu), Guangzhou, China
{wangyb, zhanglj}@lamda.nju.edu.cn
{qingguo.cqg, changgong.xz, weihua.luowh, kaifu.zkf}@alibaba-inc.com

## Abstract

Self-play fine-tuning has demonstrated promising abilities in adapting large language models (LLMs) to downstream tasks with limited real-world data. The basic principle is to iteratively refine the model with real samples and synthetic ones generated from itself. However, the existing methods primarily focus on the relative gaps between the rewards for two types of data, neglecting their absolute values. Through theoretical analysis, we identify that the gap-based methods suffer from unstable evolution, due to the potentially degenerated objectives. To address this limitation, we introduce a novel self-play fine-tuning method, namely Self-PlAy via Noise Contrastive Estimation (SPACE), which leverages noise contrastive estimation to capture the real-world data distribution. Specifically, SPACE treats synthetic samples as auxiliary components, and discriminates them from the real ones in a binary classification manner. As a result, SPACE independently optimizes the absolute reward values for each type of data, ensuring a consistently meaningful objective and thereby avoiding the instability issue. Theoretically, we show that the optimal solution of the objective in SPACE aligns with the underlying distribution of real-world data, and SPACE guarantees a provably stable convergence to the optimal distribution. Empirically, we show that SPACE significantly improves the performance of LLMs over various tasks, and outperforms supervised fine-tuning that employs much more real-world samples. Compared to gap-based self-play fine-tuning methods, SPACE exhibits remarkable superiority and stable evolution.

## 1 Introduction

In recent years, the success of large language models (LLMs) has drawn a surge of attention from the industry [2, 14, 22, 24, 47, 55, 56], and also sparked extensive investigations across various fields in academia, such as recommendation system [19, 42, 96], multimodality [4, 36, 44, 45, 48], reasoning [9, 32, 61, 89], code programming [34, 37, 39, 74], and beyond. One notable advantage of LLMs lies in their strong generalization capability, which allows them to perform well on downstream tasks via supervised fine-tuning (SFT) on real-world data annotated by human experts [38, 58].

It has been shown that effective handling of downstream tasks via SFT typically requires a large amount of annotated data, which is often impractical in real-world applications due to the expensive costs of data curation and annotation [16, 46, 82]. For this reason, self-play fine-tuning is proposed,

---

*Work done during the internship at Alibaba International Digital Commerce.
†Lijun Zhang is the corresponding author.

39th Conference on Neural Information Processing Systems (NeurIPS 2025).

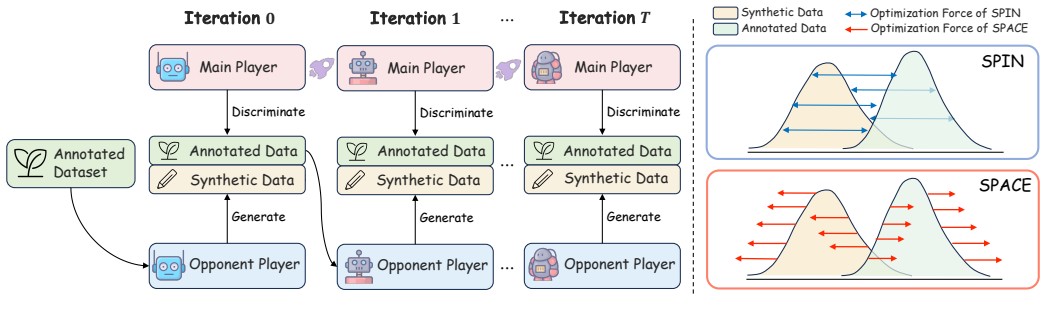

(a) The framework of self-play fine-tuning     (b) Differences between SPIN and SPACE

Figure 1: The left part (a) depicts the framework of self-play fine-tuning. The right part (b) shows the difference in optimization forces between SPIN and SPACE, where SPIN maximizes the relative gaps between annotated and synthetic data, while SPACE optimizes two types of data independently.

which takes advantage of supervision from LLMs themselves to expand the training set with synthetic samples [10, 85, 90]. Specifically, self-play fine-tuning can be viewed as a two-player game where the model competes with itself for progressive evolution. This framework is illustrated in the left part of Figure 1, where the opponent player seeks to generate synthetic data that closely resemble the annotated samples, while the main player aims to discern real data from those generated by the opponent player. It is worth noting that the main player and the opponent player refer to the same model, but instantiated with the parameters from different iterations.

Under the two-player framework, the seminal work of Chen et al. [10] introduces the first self-play fine-tuning method, termed as SPIN, whose goal is to maximize the relative gaps between the rewards of human-annotated responses $\mathbf{y} \sim p_{data}(\cdot|\mathbf{x})$ for the prompt $\mathbf{x} \sim q(\cdot)$ and those of synthetic responses $\mathbf{y}' \sim p_{\theta_t}(\cdot|\mathbf{x})$ generated from the LLM with the parameter $\theta_t$. However, it is observed that SPIN suffers unstable convergence during iterations [1, 88]. Theoretically, we attribute the instability issue to the optimization on the potentially vacuous objective. For illustration, we consider an extreme case where the generated response $\mathbf{y}'$ closely resembles the real one $\mathbf{y}$ (i.e., $\mathbf{y}' = \mathbf{y}$). In this case, the relative gap closes and thus the objective of SPIN degrades to a constant independent of $\theta$. In other words, *any* parameter

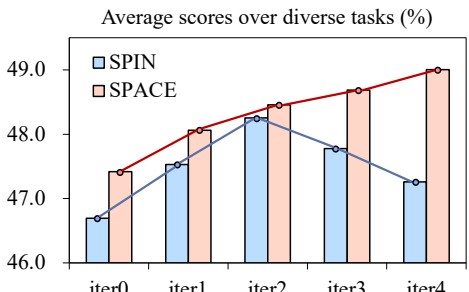

Figure 2: The average scores of SPIN and SPACE at different iterations on tasks from the HuggingFace Open LLM leaderboard.

$\theta \in \Theta$ is an optimal solution to the objective of SPIN. Our experimental results across various tasks from the HuggingFace Open LLM leaderboard reflect this phenomenon, where the performance of SPIN degrades after reaching its peak at iteration 2, as shown in Figure 2.

To resolve this limitation, we introduce a novel self-play fine-tuning method, namely Self-PlAy via Noise Contrastive Estimation (SPACE), which not only inherits the iterative evolution benefits of self-play fine-tuning but also ensures a provably *stable convergence* to the real-world data distribution. SPACE is inspired by Noise Contrastive Estimation (NCE) [26, 27], a classic technique that is commonly used in unnormalized density estimation. Specifically, SPACE incorporates the self-generated responses as auxiliary components, and distinguish real responses from synthetic ones in a binary classification manner. In contrast to SPIN [10] that focuses on the relative reward differences between real and synthetic responses, SPACE individually optimizes the absolute reward values for two types of responses. Therefore, even when the relative reward gaps diminish, the special design of SPACE ensures that the objective remains meaningful and does not collapse into a trivial constant. As a result, SPACE avoids the instability issue. The above differences are also illustrated in Figure 1(b).

Theoretically, we prove that the optimal solution of the objective in SPACE aligns with the underlying distribution $p_{data}$ of the real-world data, and SPACE ensures a provably *stable convergence* to $p_{data}$. Empirically, our results on Mistral-7B [35] and Zephyr-7B [75] show that SPACE significantly improves the average performances over different tasks from the HuggingFace Open LLM Leaderboard

[6], with up to almost 10 points increase on certain tasks, e.g., from $37.68\%$ to $46.02\%$ on GSM8K and from $23.63\%$ to $35.90\%$ on IFEval. We also show that SPACE with only $50k$ real-world responses significantly outperforms SFT with $200k$ training samples. Moreover, we compare SPACE with SPIN and other gap-based methods that are extended into the self-play fine-tuning framework. The results indicate that SPACE achieves superior performance and exhibits stable evolution during iterations.

**The main contributions.** In summary, this paper makes the following contributions: (i) We propose a novel self-play fine-tuning method SPACE, which is designed to deal with limited available expert-annotated data in downstream-task adaptation for LLMs, and address the instability issue in self-play fine-tuning; (ii) We provide theoretical guarantees that ensure the provable and stable convergence of SPACE toward the optimal distribution (cf. Theorems 2 and 3); (iii) We present results of extensive experiments that demonstrate the superiority of SPACE over existing fine-tuning strategies.

## 2    Related work

In this section, we briefly review two fine-tuning strategies for LLMs, i.e., supervised fine-tuning and self-play fine-tuning, and recent progress on noise contrastive estimation.

**Supervised fine-tuning.**    Supervised fine-tuning (SFT) is a simple and effective strategy for adapting large language models to specific tasks with supervised data [16, 82]. The goal of SFT is to learn the real-world data distribution $p_{data}(\mathbf{y}|\mathbf{x})$ where $\mathbf{x}$ denotes a prompt sampled from a task-specific distribution $q(\cdot)$, and $\mathbf{y}$ denotes the corresponding high-quality response. Formally, the loss function of SFT is formulated as:

$$\mathcal{L}_{\text{SFT}}(\theta) = -\mathbb{E}_{\mathbf{x}\sim q(\cdot), \mathbf{y}\sim p_{data}(\cdot|\mathbf{x})}\left[\log p_\theta(\mathbf{y}|\mathbf{x})\right]. \tag{1}$$

It can be verified that optimizing (1) is equivalent to minimizing the KL divergence between the distribution $p_\theta$ and the target distribution $p_{data}$, i.e., $\text{KL}(p_{data}(\cdot|\mathbf{x})||p_\theta(\cdot|\mathbf{x}))$. By the non-negativity of KL divergence, the optimal solution of (1) aligns with the real-world data distribution, i.e., $p_{\theta*}(\cdot|\mathbf{x}) = p_{data}(\cdot|\mathbf{x})$. Consequently, SFT offers a principled way to capture the underlying distribution of high-quality responses through direct supervision from human-annotated data. Nevertheless, achieving the optimal solution requires extensive high-quality data, which significantly increases the annotation costs and limits the applicability of SFT in practice.

**Self-play fine-tuning.**    Self-play framework, where the model iteratively improves the performance by competing against itself, is originally introduced in board games [68, 73], and subsequently has found its great success in many fields [5, 40, 41, 53], such as AI-player systems [70, 71] and protein discovery [78, 95]. Recently, Chen et al. [10] have introduced the self-play mechanism into large language models to address the limited high-quality data issue in SFT, and proposed the seminal self-play fine-tuning method, named SPIN, which then inspires a surge of research [11, 19, 25, 60, 65, 66, 81, 85, 88, 92]. The key idea is to enlarge the training set with synthetic data generated by the model itself, and progressively refine the model by optimizing the relative differences between the real and synthetic samples. Mathematically, at the iteration $t + 1$, SPIN aims to find a parameter $\theta_{t+1}$ satisfying

$$\theta_{t+1} = \operatorname*{argmin}_{\theta\in\Theta} \mathcal{L}_{\text{SPIN}}(\theta) = \operatorname*{argmin}_{\theta\in\Theta} \mathbb{E}\left[\ell\left(\lambda\log\frac{p_\theta(\mathbf{y}|\mathbf{x})}{p_{\theta_t}(\mathbf{y}|\mathbf{x})} - \lambda\log\frac{p_\theta(\mathbf{y}'|\mathbf{x})}{p_{\theta_t}(\mathbf{y}'|\mathbf{x})}\right)\right], \tag{2}$$

where $\ell(\cdot)$ is a monotonically decreasing convex function, and the expectation is taken over $\mathbf{x} \sim q(\cdot)$, $\mathbf{y} \sim p_{data}(\cdot|\mathbf{x})$ and $\mathbf{y}' \sim p_{\theta_t}(\cdot|\mathbf{x})$. It can be observed that with iterative updates, the capability of the model gradually improves, leading to higher-quality synthetic responses $\mathbf{y}'$. This, in turn, renders the optimization on (2) inherently unstable during iterations. In particular, when the relative gap vanishes, the objective function $\mathcal{L}_{\text{SPIN}}$ degenerates into a constant that is independent of the parameter $\theta$. This implies that *any* parameter $\theta$ in the space $\Theta$ becomes an optimal solution of $\mathcal{L}_{\text{SPIN}}$, leading to unstable optimization or even performance collapse. Recently, there have been several efforts in the literature that aim to address the unstable convergence issue of SPIN. For example, [1] employ responses from a geometric mixture of historical distributions as synthetic ones in (2) to stabilize the optimization, and [88] propose to selectively filter generated samples for continual updates. Different from them, we introduce a novel objective function in self-play fine-tuning that decouples the optimization of real and synthetic responses, and establish theoretical guarantees for the achievability and maintainability

for the underlying distribution of real-world data. As a result, our SPACE not only inherits the benefits of self-play fine-tuning, i.e., effectively fine-tuning LLMs with a small amount of annotated data, but also ensures a provable and stable convergence to the optimal distribution.

**Noise contrastive estimation.** Noise contrastive estimation (NCE) is initially proposed for density estimation in probabilistic learning [26, 27], and then has emerged as a powerful optimization technique in widespread applications recently [7, 29, 31, 33, 49, 51, 52, 54, 98]. The basic idea of NCE is to formualte the density estimation as a binary classification task, and discriminate the target distribution from an additionally introduced noise distribution. In recent years, NCE has also drawn significant attention in large language models, including preference alignment [8], information retrieval [77] and recommendation systems [87]. To the best of our knowledge, this paper is the first work that incorporates the spirit of NCE into self-play fine-tuning for large language models.

# 3 Method

In this section, we first introduce the proposed SPACE, including the update rules for main player and opponent player, and the overall objective. Then, we deliver theoretical guarantees for SPACE.

## 3.1 Our method: SPACE

Our method starts from a pretrained model $p_{\theta_0}$, which is subsequently fine-tuned on a set of high-quality data annotated for specific downstream tasks. Our SPACE is built upon the self-play framework, where a main player and an opponent player compete with each other for progressive evolution. In this framework, both the main player and the opponent player are the same model, but equipped with different parameters. Specifically, at each iteration $t + 1$, the opponent player parameterized by $\theta_t$ generates the synthetic response $\mathbf{y}'$ for a prompt $\mathbf{x}$ by sampling from the distribution $p_{\theta_t}(\cdot|\mathbf{x})$. Then, the main player takes the human-annotated response $\mathbf{y}$ and the synthetic one $\mathbf{y}'$ as inputs and updates its parameters to obtain $\theta_{t+1}$. Subsequently, the opponent player adjusts its parameters so that it can generate synthetic responses for the next iteration. Overall, this process includes two steps: the updates for the main player and the opponent player, which are separately detailed in the following.

**Updates for the main player.** Recall that the goal of the main player is to discriminate the human-annotated responses and the synthetic ones. Drawing inspiration from noise contrastive estimation [26, 27], we formulate the discrimination as a binary classification problem, where the main player serves as a classifier to distinguish between the human-annotated responses $\mathbf{y}$ and the synthetic responses $\mathbf{y}'$. This formulation shares the same spirit with differentiation theory in cognitive development [21], where learning emerges through progressive extraction of meaningful patterns from environments. Formally, we first employ the commonly used log ratio [62] as the reward for a response $\mathbf{u}$: $r(\mathbf{u}|\mathbf{x}) = \log p_\theta(\mathbf{u}|\mathbf{x}) - \log p_{\hat{\theta}_t}(\mathbf{u}|\mathbf{x})$, where $\theta$ and $\hat{\theta}_t$ denote parameters of main player and opponent player, respectively. Then, we establish the relationship between the rewards for two types of responses (i.e., $\mathbf{y}$ and $\mathbf{y}'$), and their corresponding posterior probabilities, as shown below.

**Proposition 1.** *Let $p_{\theta^*}$ be the real data distribution, and $r^*(\cdot|\mathbf{x}) = \log p_{\theta^*}(\cdot|\mathbf{x}) - \log p_{\hat{\theta}_t}(\cdot|\mathbf{x})$ be the reward for $p_{\theta^*}$. Then, given a mixture distribution $p_{mix}(\mathbf{y}|\mathbf{x}) = (1 + \mu)^{-1} p_{\theta^*}(\mathbf{y}|\mathbf{x}) + \mu(1 + \mu)^{-1} p_{\hat{\theta}_t}(\mathbf{y}|\mathbf{x})$ with the ratio $\mu$, the posterior probabilities of a sample $(\mathbf{x}, \mathbf{y})$ from $p_{mix}$ are*

$$p_{\theta^*}(c = 1|\mathbf{y}, \mathbf{x}) = \frac{1}{1 + \mu \exp(-r^*(\mathbf{y}|\mathbf{x}))}, \quad p_{\theta^*}(c = 0|\mathbf{y}, \mathbf{x}) = \frac{1}{1 + \mu^{-1} \exp(r^*(\mathbf{y}|\mathbf{x}))} \quad (3)$$

*where $c$ is the label indicating whether $\mathbf{y}$ is real ($c = 1$) or synthetic ($c = 0$).*

The above proposition reveals that the optimal reward captures the posterior probability of a response, thereby establishing a principled connection between the reward function and binary classification over a mixture distribution. To be precise, a higher reward $r(\mathbf{y}|\mathbf{x})$ corresponds to a higher probability $p_\theta(c = 1|\mathbf{y}, \mathbf{x})$, indicating that the response $\mathbf{y}$ is more likely to be real; conversely, a lower reward implies a higher probability $p_\theta(c = 0|\mathbf{y}, \mathbf{x})$ and a less realistic response. Therefore, with Proposition 1, we can conveniently leverage the maximum likelihood strategy to train the main player to find a parameter $\theta_{t+1}$ that can distinguish between real and synthetic samples:

$$\theta_{t+1} = \underset{\theta \in \Theta}{\operatorname{argmax}} \, \mathbb{E} \left[ \log p_\theta(c = 1|\mathbf{y}, \mathbf{x}) + \mu \log p_\theta(c = 0|\mathbf{y}', \mathbf{x}) \right], \quad (4)$$

---

**Algorithm 1** Self-PlAy via Noise Contrastive Estimation (SPACE)

---

**Inputs**: Annotated set $\{\mathbf{x}_i, \mathbf{y}_i\}_{i=1}^n$, the generation ratio $\mu$, and a pretrained LLM $p_{\theta_0}$
**Initialization**: Set $p_{\hat{\theta}_0} = p_{\theta_0}$, and compute the size of synthetic data $m = \mu n$

1: **for** $t = 0, 1, 2, \cdots$ **do**
2:      Generate the synthetic tuples $\{\mathbf{x}_j, \mathbf{y}_j'\}_{j=1}^m$ by the opponent player $p_{\theta_t}$, i.e., $\mathbf{y}_j' \sim p_{\theta_t}(\cdot|\mathbf{x}_j)$
3:      Obtain $p_{\theta_{t+1}}$ by minimizing the loss function (8)
4: **end for**

---

where the expectation is taken over the distributions $\mathbf{x} \sim q(\cdot), \mathbf{y} \sim p_{data}(\cdot|\mathbf{x})$ and $\mathbf{y}' \sim p_{\hat{\theta}_t}(\cdot|\mathbf{x})$. Substituting (3) into (4) delivers

$$\theta_{t+1} = -\underset{\theta \in \Theta}{\operatorname{argmin}} \, \mathbb{E}\left[\log \sigma_\mu\left(\log \frac{p_\theta(\mathbf{y}|\mathbf{x})}{p_{\hat{\theta}_t}(\mathbf{y}|\mathbf{x})}\right) + \mu \log \sigma_{\mu^{-1}}\left(\log \frac{p_{\hat{\theta}_t}(\mathbf{y}'|\mathbf{x})}{p_\theta(\mathbf{y}'|\mathbf{x})}\right)\right] \tag{5}$$

where $\sigma_\mu(x) = (1 + \mu \exp(-x))^{-1}$. Compared to the objective of SPIN in (2), (5) separately optimizes the rewards of the real responses and the synthetic ones. The separate structure ensures that, when $\mathbf{y}'$ closely resembles $\mathbf{y}$, (5) does not degenerate into a constant independent of $\theta$. As a result, optimizing (5) guarantees a provably stable convergence, which will be theoretically elaborated later.

**Updates for the opponent player.** Given the newly obtained main player $p_{\theta_{t+1}}$, we proceed to update the opponent player, the goal of which is to generate high-quality synthetic responses that can mislead the main player. To achieve this, a natural intuition is to adjust the opponent player by maximizing the reward associated with synthetic responses, which is formulated as follows:

$$\hat{\theta}_{t+1} = \underset{\hat{\theta} \in \Theta}{\operatorname{argmax}} \, \mathbb{E}\left[r(\mathbf{y}'|\mathbf{x})\right] = \underset{\hat{\theta} \in \Theta}{\operatorname{argmax}} \, \mathbb{E}\left[\log p_{\theta_{t+1}}(\mathbf{y}'|\mathbf{x}) - \log p_{\hat{\theta}}(\mathbf{y}'|\mathbf{x})\right] \tag{6}$$

where the expectation is taken over the distributions $\mathbf{x} \sim q(\cdot)$ and $\mathbf{y}' \sim p_{\hat{\theta}}(\cdot|\mathbf{x})$. Mathematically, (6) measures the KL divergence between $p_{\hat{\theta}}$ and $p_{\theta_{t+1}}$. Consequently, by the non-negativity of KL divergence, we can obtain that the closed-form solution of (6) takes the form of

$$p_{\hat{\theta}_{t+1}}(\mathbf{y}|\mathbf{x}) = p_{\theta_{t+1}}(\mathbf{y}|\mathbf{x}). \tag{7}$$

The above finding is particularly appealing, as it indicates that the updated parameter of the main player coincide exactly with the optimal solution for the opponent player. Consequently, we can directly set the parameter of the opponent player for the next iteration to be the same as that of the main player, i.e., $p_{\hat{\theta}_{t+1}} = p_{\theta_{t+1}}$, which inherently reveals the *self-play* nature of our SPACE.

**The overall objective.** Now, we are ready to introduce the overall objective of SPACE, by combining the above two update steps together. In details, we substitute (7) into (5), and deliver the following objective function:

$$\mathcal{L}_{\text{SPACE}}(\theta) = -\mathbb{E}\left[\log \sigma_\mu\left(\log \frac{p_\theta(\mathbf{y}|\mathbf{x})}{p_{\theta_t}(\mathbf{y}|\mathbf{x})}\right) + \mu \log \sigma_{\mu^{-1}}\left(\log \frac{p_{\theta_t}(\mathbf{y}'|\mathbf{x})}{p_\theta(\mathbf{y}'|\mathbf{x})}\right)\right]. \tag{8}$$

We summarize the detailed procedure of SPACE in Algorithm 1. Specifically, at the begining, we initialize the opponent player with the pretained LLM, i.e., $p_{\hat{\theta}_0} = p_{\theta_0}$, and compute the size of synthetic data $m = \mu n$. At each iteration $t$, the opponent player generates the synthetic response $\mathbf{y}' \sim p_{\hat{\theta}_t}(\cdot|\mathbf{x})$ for each prompt $\mathbf{x}$. Then, the main player takes annotated and synthetic data as inputs, and obtains $p_{\theta_{t+1}}$ by minimizing (8). Subsequently, the opponent player copys $\theta_{t+1}$ as its parameter according to (7), to produce new synthetic responses for the next iteration.

**In-depth discussions.** In the following, we analyze the underlying cause of instability issue in SPIN, and explain how SPACE addresses this issue and achieves stable convergence. As shown in (2), SPIN aims to optimize the gaps bewteen the annotated responses and synthetic ones. In fact, the gap-based structure is vulnerable for self-play fine-tuning. Specifically, with the refinement of synthetic responses, the gaps between real and self-generated data gradually close, and the gap-based objective will degenerate to a trivial constant when the gap vanishes. In this case, any solution in the parameter space can be considered optimal, leading to the instability issue. Moreover, if existing gap-based methods for large language models, such as IPO [3] and SimPO [50], are extended to self-play fine-tuning scenarios, they are similarly suffer from the instability issue, due to the inherent

weakness of gap-based objectives. More detailed introduction about gap-based extensions can be found in Appendix B. In contrast, our proposed SPACE circumvents this issue by decoupling the optimization of real and synthetic responses: instead of maximizing their relative difference, SPACE independently optimizes the absolute reward value for each, thereby maintaining a stable evolution.

## 3.2 Theoretical analysis

In the following, we provide theoretical analysis of SPACE to understand its underlying principles. We begin by presenting the gradient analysis for the objective function (8) in SPACE.

**Theorem 1.** *The gradient of* (8) *for a response* $\mathbf{u}$ *takes the form of*

$$\nabla_\theta \mathcal{L}_{\texttt{SPACE}}(\theta) = -\mathbb{E}_{\mathbf{x} \sim q(\cdot)} \left[ \sigma_{\mu^{-1}}(-r(\mathbf{x}, \mathbf{u}))(p_{data}(\mathbf{u}|\mathbf{x}) - p_{\theta_t}(\mathbf{u}|\mathbf{x}))\nabla_\theta \log p_\theta(\mathbf{u}|\mathbf{x}) \right]. \quad (9)$$

**Remark.** The above theorem indicates that the update rules in SPACE enjoy a desirable "response-dependent" property, i.e., the likelihood of a response $\mathbf{u}$ is adjusted based on itself. Specifically, we can examine this property by considering two types of responses: $\mathbf{y} \sim p_{data}(\cdot|\mathbf{x})$ sampled from the real-world distribution $p_{data}$ and $\mathbf{y}' \sim p_{\theta_t}(\cdot|\mathbf{x})$ generated by the model $p_{\theta_t}$. Note that $\sigma_{\mu^{-1}}(\cdot) > 0$ holds always true for $\mu > 0$. Therefore, the former case where $p_{data}(\mathbf{y}|\mathbf{x}) - p_{\theta_t}(\mathbf{y}|\mathbf{x}) \geq 0$ leads to an increase in the log-probability of $\mathbf{y}$. For the latter case where $p_{data}(\mathbf{y}'|\mathbf{x}) - p_{\theta_t}(\mathbf{y}'|\mathbf{x}) \leq 0$, the log-probability of $\mathbf{y}'$ will be decreased. This favorable property ensures that SPACE can effectively distinguish annotated responses and synthetic responses by adaptively adjusting their probabilities.

**Remark.** Note that although (9) shares a similar structure with the policy gradient in reinforcement learning (RL), our SPACE exhibits fundamental differences from reinforcement learning methods. Specifically, in the standard policy gradient formulation (e.g., REINFORCE [84]), the gradient takes the form of $\nabla_\theta \mathcal{L}(\theta) = -\mathbb{E}_{\mathbf{u} \sim p_\theta}[R(\mathbf{u}|\mathbf{x}) \cdot \nabla_\theta \ln p_\theta(\mathbf{u}|\mathbf{x})]$, where $R(\mathbf{u}|\mathbf{x})$ denotes the reward associated with response $\mathbf{u}$. Superficially, (9) can be viewed as a special case of this formulation by choosing $R(\mathbf{u}|\mathbf{x}) = \sigma_{\mu^{-1}}(-r(\mathbf{x}, \mathbf{u}))(p_{data}(\mathbf{u}|\mathbf{x}) - p_{\theta_t}(\mathbf{u}|\mathbf{x}))$, but it is important to note that SPACE is based on the self-play fine-tuning framework, where the goal is to iteratively improve performances through two-player competition. This fundamentally differs from the exploration-exploitation trade-off in RL. Additionally, SPACE optimizes (8) in a supervised-learning manner, without involving explicit rewarding or rollout phrases, which are essential components of standard RL methods.

Then, we demonstrate that SPACE enjoys the *achievability* and *maintainability* for the annotated data distribution $p_{data}$, as established in the following two theorems.

**Theorem 2.** *By minimizing the loss function* (8)*,* SPACE *is able to capture the annotated data distribution* $p_{data}$*, i.e.,* $p_{\theta^*} = p_{data}$*, where* $\theta^*$ *denotes the optimal solution to* (8)*.*

**Theorem 3.** *Suppose* $p_{\theta_t}$ *has already converged to the annotated data distribution* $p_{data}$ *at iteration* $t$*, i.e.,* $p_{\theta_t}(\cdot|\mathbf{x}) = p_{data}(\cdot|\mathbf{x})$*. Then, at iteration* $t+1$*,* SPACE *still ensures* $p_{\theta_{t+1}}(\cdot|\mathbf{x}) = p_{data}(\cdot|\mathbf{x})$*.*

**Remark.** Theorem 2 establishes the *achievability* for the target distribution $p_{data}$, i.e., it guarantees that minimizing the objective function (8) yields a distribution that aligns with $p_{data}$. This result provides theoretical support for SPACE, demonstrating its ability to align with the target distribution, which is particularly important in fine-tuning LLM for downstream tasks.

**Remark.** Theorem 3 further emphasizes the *maintainability* for the target distribution $p_{data}$, which can be regarded as a complement to the achievability property in Theorem 2, highlighting the long-term stability for $p_{data}$. Notably, this stable convergence property is also critical for self-play fine-tuning methods, as it ensures that the iterative optimization process retains the optimal solution without divergence, thereby mitigating the instability issue during iterations. This appealing property theoretically exhibits the advantages of SPACE over SPIN [10].

## 4 Experiments

In this section, we present empirical studies to validate the effectiveness of SPACE. We first describe experimental settings, including datasets, pre-trained models, implementations and evaluations. We then report the results with corresponding analyses. More experiments are provided in Appendix C.

Table 1: Performance (%) comparisons on various tasks among our SPACE (red) , SPIN, S-IPO and S-SimPO. *Avg* denotes the average score over different tasks, where highest and second-highest scores over iterations 0 to 4 are highlighted in **bold** and underline, respectively.

| | Model | ARC | GSM8K | HellaSwag | MMLU | TruthfulQA | Winogrande | IFEval | BBH | GPQA | MMLU$_{pro}$ | Avg |
|---|---|---|---|---|---|---|---|---|---|---|---|---|
| | Mistral-7B | 61.01 | 37.68 | 83.24 | 57.86 | 42.62 | 74.03 | 23.63 | 44.26 | 29.86 | 29.99 | 48.42 |
| S-IPO | Iter0 | 60.49 | 40.96 | 83.58 | 57.17 | 45.45 | 73.88 | 20.85 | 42.33 | 30.71 | 30.01 | **48.54** |
| | Iter1 | 60.07 | 38.91 | 83.47 | 57.29 | 43.62 | 73.80 | 21.98 | 42.08 | 30.19 | 29.90 | 48.13 |
| | Iter2 | 60.07 | 39.35 | 83.19 | 57.29 | 43.35 | 73.80 | 23.29 | 41.69 | 30.50 | 29.86 | 48.24 |
| | Iter3 | 60.67 | 41.05 | 83.42 | 57.03 | 44.35 | 74.27 | 23.00 | 41.73 | 29.32 | 29.69 | 48.45 |
| | Iter4 | 61.43 | 40.11 | 83.56 | 56.61 | 45.56 | 74.11 | 23.11 | 41.05 | 29.31 | 29.36 | 48.42 |
| S-SimPO | Iter0 | 61.35 | 39.67 | 83.76 | 57.29 | 47.44 | 74.66 | 24.88 | 41.86 | 30.10 | 30.04 | 49.11 |
| | Iter1 | 61.69 | 41.51 | 83.77 | 57.48 | 48.98 | 74.90 | 23.01 | 41.86 | 29.90 | 30.22 | 49.33 |
| | Iter2 | 61.52 | 37.09 | 83.77 | 57.16 | 49.58 | 74.51 | 22.44 | 43.04 | 29.17 | 29.97 | 48.83 |
| | Iter3 | 61.26 | 35.22 | 83.70 | 57.35 | 49.88 | 74.82 | 21.15 | 42.80 | 29.04 | 30.01 | 48.52 |
| | Iter4 | 61.35 | 34.38 | 83.68 | 57.48 | 50.02 | 74.19 | 21.31 | 42.67 | 29.16 | 39.40 | **49.36** |
| SPIN | Iter0 | 61.18 | 38.29 | 83.49 | 57.88 | 43.73 | 74.11 | 22.09 | 44.60 | 29.62 | 30.15 | 48.51 |
| | Iter1 | 61.52 | 32.85 | 84.02 | 57.32 | 47.69 | 73.95 | 21.92 | 40.59 | 28.22 | 30.12 | 47.82 |
| | Iter2 | 62.20 | 40.26 | 83.60 | 58.03 | 46.92 | 74.43 | 25.06 | 43.50 | 28.95 | 30.37 | **49.33** |
| | Iter3 | 62.29 | 34.87 | 84.01 | 58.09 | 45.72 | 75.14 | 24.64 | 43.03 | 28.68 | 29.62 | 48.61 |
| | Iter4 | 61.86 | 34.70 | 83.99 | 57.99 | 46.00 | 75.14 | 25.73 | 43.34 | 27.10 | 29.82 | 48.57 |
| SPACE (ours) | Iter0 | 62.71 | 41.32 | 83.79 | 58.67 | 47.15 | 74.90 | 26.96 | 45.87 | 29.66 | 30.73 | 50.18 |
| | Iter1 | 64.85 | 45.41 | 83.86 | 58.87 | 48.99 | 75.14 | 31.51 | 45.73 | 29.46 | 30.94 | 51.48 |
| | Iter2 | 65.78 | 45.55 | 84.25 | 58.57 | 50.71 | 73.95 | 33.01 | 45.38 | 29.52 | 31.59 | 51.83 |
| | Iter3 | 65.96 | 45.84 | 84.39 | 58.53 | 51.34 | 74.19 | 33.19 | 45.38 | 29.90 | 31.44 | 52.03 |
| | Iter4 | 65.87 | 46.02 | 84.44 | 58.50 | 51.86 | 74.51 | 35.90 | 45.28 | 30.50 | 31.41 | **52.43** |

Figure 3: The performance comparisons among four self-play fine-tuning methods on Mistral-7B. (a) the average scores over different tasks; (b) the performances on GSM8K; (c) the average ranks over different iterations, where the best rank among iterations 0 to 4 is highlighted with a "gold medal".

## 4.1 Experimental settings

Following Chen et al. [10], we randomly sample $50k$ prompts with their corresponding high-quality responses from the Ultrachat200k dataset [15], and choose Zephyr-7B-SFT-full [75] and Mistral-7B-Base [35] as pretrained models in experiments. During the training, we first generate synthetic response $\mathbf{y}'$ for each $\mathbf{x}$ with the latest model at each iteration. The resulting synthetic response is then combined with annotated one to update the large language model for the subsequent iteration.

We evaluate the performances with different tasks from the HuggingFace Open LLM Leaderboard [6, 18], each targeting a distinct capability of LLMs. These tasks cover a range of domains: science question answering with ARC-Challenge [12] and GPQA [64], mathematical reasoning with GSM8K [13], commonsense inference with Winogrande [67] and HellaSwag [93], multitask language understanding through MMLU [30] and MMLU-Pro [83], truthfulness and factuality with TruthfulQA [43], instruction following using IFEval [97], and complex reasoning with BBH [72]. All tasks are implemented with the default configurations provided by the Language Model Evaluation Harness [20]. Our implementation is based on the codebase Alignment Handbook [76] and the Accelerate library [23]. We choose RMSProp [69] with default configurations as the optimizer, and set the

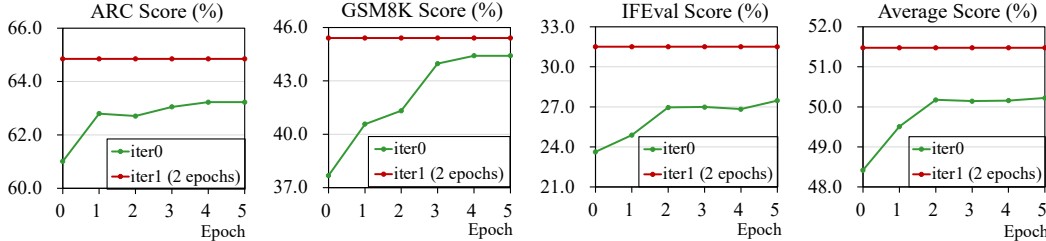

Figure 4: Performance comparisons between the model trained with multiple epochs at iteration 0 (green) and that trained with two epochs at iteration 1 (red).

global batch size as 64 and the epoch as 2. For SPACE, we choose to set the generation ratio $\mu = 1$ (i.e., $m = n$) in our experiments, and we will return to this configuration later.

## 4.2 Experimental results

**Comparison to self-play fine-tuning baselines.** We compare SPACE with SPIN [10], as well as self-play variants of two gap-based methods, namely S-IPO from IPO [3] and S-SimPO from SimPO [50]. The detailed descriptions for two self-play variant methods are provided in Appendix B.

The experimental results on Mistral-7B are shown in Table 1, and we defer the results on Zephyr-7B to Appendix C. The results from Table 1 indicate that all self-play methods surpass the initial base model with notable improvements. Specifically, our SPACE improves the average score of Mistral-7B from 48.42% to 52.43%. Remarkably, it achieves substantial performance gains on GSM8K and IFEval with almost 10 points improvements. Additionally, compared to other self-play baselines, our SPACE demonstrates superior performance and stable evolution. For clarity, we present the average scores over different tasks for each method in Figure 3(a). We observe two key findings: (i) the performance improvements predominantly occur within the first two iterations, which aligns with the conclusions of Chen et al. [10]; (ii) SPIN as well as the self-play variants of IPO and SimPO suffers degradation after reaching their performance peak. Specifically, for the GSM8K task shown in Figure 3(b), SPIN, S-IPO and S-SimPO show improvements at the initial iterations but incurs performance decline after that. In contrast, our SPACE maintains its stable improvements over iterations. Moreover, we also compare the average ranks over iterations, as shown in Figure 3(c). The results indicate that three baselines achieve their best performance in early iterations, whereas SPACE maintains steady improvements over iterations, ultimately achieving the best rank at the final iteration.

**Comparison to training with more epochs.** Then, we investigate the effectiveness of the self-play mechanism in SPACE. We conduct experiments on Mistral-7B by comparing the model trained for multiple epochs at iteration 0 with the *fixed* synthetic responses, versus the model trained on *re-generated* synthetic responses at iteration 1 with two epochs. The results are illustrated in Figure 4. We observe that training with multiple epochs at iteration 0 initially improves performance but gradually plateaus, and fails to surpass the performance achieved at iteration 1. We consider the performance bottleneck primarily to the stagnation of negative samples. Specifically, at iteration 0, despite minor performance improvements, the model undergoes multiple rounds of training on an *unchanged* dataset. In contrast, at iteration 1, we first refine the training data by leveraging the self-play mechanism to regenerate synthetic samples with the latest model, and then train the model on the enhanced data, delivering substantial performance gains. This observation demonstrates that improving the quality of synthetic samples plays a critical role in enhancing model performance, validating the effectiveness of the self-play mechanism in our SPACE.

**The generation ratio.** Next, we examine the impact of the generation ratio $\mu$, which quantifies the proportion of synthetic responses relative to human-annotated ones, on the performance of SPACE. Specifically, we conduct experiments on the Mistral-7B with different generation ratios from $\{1.0, 3.0, 7.0\}$. For $\mu = 1.0$, we generate $50k$ synthetic responses for the $50k$ human-annotated responses in the training dataset at each iteration. For $\mu = 3.0$ and $\mu = 7.0$, we generate $150k$ and $350k$ synthetic responses respectively, while maintaining the same $50k$ human-annotated responses. The experimental results are shown in Table 2. Overall, increasing the number of generated samples can improve the performance of SPACE, though this improvement gradually diminishes in later iterations. Additionally, it is important to note that such performance gains come at the cost of significantly higher computational resources. For instance, when $\mu = 3.0$, the generation phase

Table 2: Performance (%) comparisons with different generation ratios of SPACE accross iterations. *Avg* denotes the average score over multiple tasks, and *Gen Ratio* denotes the generation ratio.

| Gen Ratio | | ARC | GSM8K | HellaSwag | MMLU | TruthfulQA | Winogrande | IFEval | BBH | GPQA | MMLU$_{pro}$ | Avg |
|---|---|---|---|---|---|---|---|---|---|---|---|---|
| Mistral-7B | | 61.01 | 37.68 | 83.24 | 57.86 | 42.62 | 74.03 | 23.63 | 44.26 | 29.86 | 29.99 | 48.42 |
| Iter 0 | 1.0 | 62.71 | 41.32 | 83.79 | 58.67 | 47.15 | 74.90 | 26.96 | 45.87 | 29.66 | 30.73 | 50.18 |
| | 3.0 | 62.97 | 42.34 | 83.25 | 58.70 | 47.52 | 74.98 | 36.05 | 47.00 | 29.27 | 30.39 | 51.25 |
| | 7.0 | 63.48 | 41.26 | 82.79 | 58.55 | 46.65 | 74.90 | 36.93 | 46.37 | 29.78 | 29.85 | 51.06 |
| Iter 1 | 1.0 | 64.85 | 45.41 | 83.86 | 58.87 | 48.99 | 75.14 | 31.51 | 45.73 | 29.46 | 30.94 | 51.48 |
| | 3.0 | 65.10 | 43.03 | 83.74 | 58.71 | 50.95 | 73.88 | 36.44 | 46.24 | 29.65 | 30.31 | 51.80 |
| | 7.0 | 64.99 | 41.32 | 83.37 | 58.49 | 48.37 | 73.32 | 40.64 | 46.61 | 29.33 | 30.29 | 51.67 |
| Iter 2 | 1.0 | 65.78 | 45.55 | 84.25 | 58.57 | 50.71 | 73.95 | 33.01 | 45.38 | 29.52 | 31.59 | 51.83 |
| | 3.0 | 64.76 | 41.39 | 83.67 | 58.51 | 48.89 | 73.24 | 41.93 | 46.20 | 29.20 | 30.48 | 51.83 |
| | 7.0 | 64.59 | 41.49 | 83.63 | 58.50 | 49.23 | 73.56 | 40.33 | 46.22 | 29.69 | 30.29 | 51.75 |

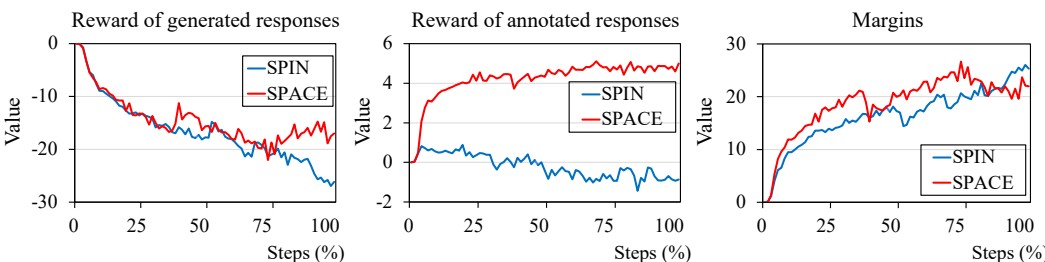

Figure 6: Comparison of rewards for generated and human-annotated responses, and the corresponding gap between two types of responses, for SPIN and SPACE at iteration 0.

requires $4.50$ hours and the training phase requires $6.80$ hours; when $\mu = 7.0$, these increase to $6.20$ hours and $12.23$ hours respectively, substantially *higher* than the computational costs for $\mu = 1.0$ ($0.83$ hours for generation and $3.20$ hours for training). Because of the substantial computational costs, we choose to use $\mu = 1.0$ rather than larger generation ratios in our experiments.

**Comparison to SFT.** In this part, we compare SPACE with SFT to demonstrate that SPACE with less human-annotated data can achieve better performance than SFT that utilizes much more data.

The experiments are conducted on Mistral-7B [35] with varying amounts of human-annotated data. Specifically, we train SPACE on nested subsets of $12k$, $25k$, and $50k$ samples from Ultrachat200k [15], with each larger subset encompassing the smaller ones. Since the performance improvements primarily occur within the first two iterations, we train SPACE for two iterations on each subset. For SFT, we train the model on the a subsets of $100k$ samples from Ultrachat200k, the complete dataset with $200k$ data, and an expanded version with $400k$ samples where each sample is duplicated once. We display the average scores of SPACE and SFT in Figure 5. It is observed that both SPACE and SFT significantly improve model performance compared to the initial base model. However, as the amount of training data increases from 100k to 400k, the performance gains of SFT gradually reach a plateau, suggesting that additional human-annotated responses

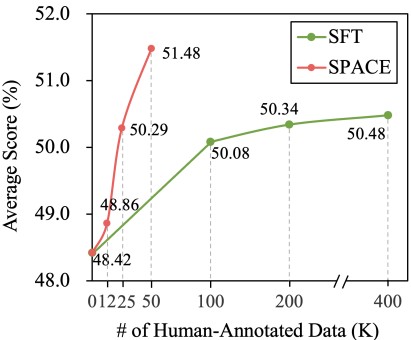

Figure 5: The average score with different sizes of annotated data. The start point denotes the performance of base model.

provides limited benefit for further performance improvement. In contrast, SPACE achieves superior performance with fewer human-annotated responses, compared to SFT trained on a much larger dataset, demonstrating its effectiveness in scenarios with limited human supervision.

**Computational costs.** In SPACE, the main computational costs come from the generation of synthetic responses and the model training. In implementations, we follow Chen et al. [10] utilizing the

Table 3: The computation time of generation and training for different iterations. *Gen* and *Train* denote the generation and training phases, respectively.

| Iteration | Iter 0 | | Iter 1 | | Iter 2 | | Iter 3 | | Iter 4 | |
|---|---|---|---|---|---|---|---|---|---|---|
| Phase | Gen | Train | Gen | Train | Gen | Train | Gen | Train | Gen | Train |
| Time | 0.83h | 3.20h | 0.83h | 3.20h | 0.83h | 3.20h | 0.83h | 3.20h | 0.83h | 3.20h |

Accelerate library [23] to generate synthetic responses in a distributed manner with the global batch size 256. All experiments are conducted on a single machine equipped with 8 H100 GPUs, and we report the costs of generation and training in Table 3. During the generation phase, we produce $50k$ synthetic responses for the input prompts and suffer the time cost of $0.83$ hours per iteration. During the training phase, we train the model with 2 epochs with the total of $3.20$ hours per iteration. It is observed that the majority of computational costs are concentrated in the training phase.

**Preventing from reward decline for human-annotated responses.** Finally, we discuss the reward decline issue for high-quality responses, which have raised wide investigations in preference alignments [17, 59, 63, 86], but has not yet been discussed in self-play fine-tuning. According to (2), SPIN aims to maximize the relative gap between the rewards of annotated response and self-generated one. In experiments, we observe that the reward margins between real and synthetic data increase (right of Figure 6), but this improvement comes from the different rates of decline between the two types of responses. In fact, the decreasing reward for human-annotated responses (middle of Figure 6) is undesirable, as it explicitly reflects the decline in the priority of human-annotated responses. In other words, the capability to generate high-quality responses gradually deteriorates [91].

It is worth noting that SPACE avoids the reward decline for real responses, and exhibits an increasing trend (middle of Figure 6). This can be attributed to the inherent structure of the objective function in (8), which explicitly increases the reward for human-annotated responses while decreasing it for self-generated ones. Moreover, the favorable *response-dependent* property (Theorem 1) also shows that SPACE can naturally support distinct update trends for two types of responses.

## 5   Conclusion and future work

In this paper, we investigate self-play fine-tuning for large language models, and introduce a novel method named SPACE to resolve the instability issue of gap-based methods. The pivotal idea of SPACE is to treat synthetic samples as auxiliary components and distinguish them from real ones in a binary classification manner. In this way, SPACE is able to optimize the absolute reward values for annotated and synthetic samples independently, avoiding the unstable convergence. Theoretically, we show that SPACE enjoys the favorable properties, i.e., the *achievability* and *maintainability* for the real-world data distribution $p_{data}$. In other words, SPACE can provably converge to $p_{data}$, and maintain it during iterations, ensuring a stable evolution. Empirically, extensive experiments demonstrate that SPACE not only outperforms SPIN and other gap-based self-play baselines, but also achieves superior performance to SFT, while utilizing substantially less annotated data.

There are many directions for future research. First, in existing self-play fine-tuning methods, the total number of iterations is a hyper-parameter that has to be configured in advance, which may lead to unnecessary computation or premature convergence. In the future, we will investigate self-play fine-tuning methods that can adaptively adjust the number of iterations based on the evolution progress. Second, the current methods including SPIN [10] and our SPACE are designed for the scenario where the optimal distribution is fixed during iterations, which, however, is not necessarily satisfied in practice. Therefore, it is also interesting to explore self-play fine-tuning in the non-stationary scenario, which may require the advanced techniques from online optimization [28, 57, 79, 80, 94]. Third, we also consider to apply SPACE to complex scenarios with limited annotated data, such as LLM agents.

## Acknowledgments and Disclosure of Funding

This work was partially supported by NSFC (U23A20382), and the Collaborative Innovation Center of Novel Software Technology and Industrialization.

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

# A Theoretical proofs

In this section, we provide the proofs of the proposition and theorems presented in the main paper.

## A.1 Proof of Proposition 1

For the human-annotated data, each response is drawn from the real-world distribution $p_{data}(\cdot|\mathbf{x})$; whereas for the synthetic data, each response comes from the opponent player $p_{\hat{\theta}_t}(\cdot|\mathbf{x})$. Consequently, their conditional probabilities are

$$p(\mathbf{y}|c = 1, \mathbf{x}) = p_{data}(\mathbf{y}|\mathbf{x}) = p_{\theta^*}(\mathbf{y}|\mathbf{x}) \text{ and } p(\mathbf{y}'|c = 0, \mathbf{x}) = p_{\hat{\theta}_t}(\mathbf{y}'|\mathbf{x}),$$

where $c$ is the label indicating the real or synthetic, and $\theta^*$ is the optimal parameters for LLM. Then, given the mixture distribution

$$p_{mix}(\mathbf{y}|\mathbf{x}) = \frac{1}{1 + \mu}p_{\theta^*}(\mathbf{y}|\mathbf{x}) + \frac{\mu}{1 + \mu}p_{\hat{\theta}_t}(\mathbf{y}|\mathbf{x})$$

we can calculate the posterior probabilities of sampling a response from $p_{mix}$ using Bayes' theorem:

$$p(c = 1|\mathbf{y}, \mathbf{x}) = \frac{p_{\theta^*}(\mathbf{y}|\mathbf{x})}{p_{\theta^*}(\mathbf{y}|\mathbf{x}) + \mu p_{\hat{\theta}_t}(\mathbf{y}|\mathbf{x})} \text{ and } p(c = 0|\mathbf{y}', \mathbf{x}) = \frac{\mu p_{\hat{\theta}_t}(\mathbf{y}'|\mathbf{x})}{p_{\theta^*}(\mathbf{y}'|\mathbf{x}) + \mu p_{\hat{\theta}_t}(\mathbf{y}'|\mathbf{x})}.$$

Finally, substituting the implicit reward $r^*(\mathbf{y}|\mathbf{x}) = \log p_{\theta^*}(\mathbf{y}|\mathbf{x}) - \log p_{\hat{\theta}_t}(\mathbf{y}|\mathbf{x})$ into the posterior probabilities completes the proof.

## A.2 Proof of Theorem 1

Recall that (8) can be rewritten as

$$\mathcal{L}_{\text{SPACE}}(\theta) = -\mathbb{E}\left[\int p_{data}(\mathbf{y}|\mathbf{x}) \log\left(\frac{p_\theta(\mathbf{y}|\mathbf{x})}{\tilde{p}_{\theta,\theta_t}(\mathbf{y}|\mathbf{x})}\right) + \mu p_{\theta_t}(\mathbf{y}|\mathbf{x}) \log\left(\frac{\mu p_{\theta_t}(\mathbf{y}|\mathbf{x})}{\tilde{p}_{\theta,\theta_t}(\mathbf{y}|\mathbf{x})}\right) d\mathbf{y}\right],$$

where the expectation is taken over $\mathbf{x} \sim q(\cdot)$, and $\tilde{p}_{\theta,\theta_t}(\mathbf{y}|\mathbf{x}) \triangleq p_\theta(\mathbf{y}|\mathbf{x}) + \mu p_{\theta_t}(\mathbf{y}|\mathbf{x})$. Then, we take the derivative of $\mathcal{L}_{\text{SPACE}}(\theta)$ with respect to $p_\theta$ as below:

$$\nabla_\theta \mathcal{L}_{\text{SPACE}}(\theta) = -\nabla_\theta \mathbb{E}\left[\int p_{data}(\mathbf{y}|\mathbf{x}) \log\left(\frac{p_\theta(\mathbf{y}|\mathbf{x})}{\tilde{p}_{\theta,\theta_t}(\mathbf{y}|\mathbf{x})}\right) + \mu p_{\theta_t}(\mathbf{y}|\mathbf{x}) \log\left(\frac{\mu p_{\theta_t}(\mathbf{y}|\mathbf{x})}{\tilde{p}_{\theta,\theta_t}(\mathbf{y}|\mathbf{x})}\right) d\mathbf{y}\right]$$

$$= -\mathbb{E}\left[\frac{p_{data}(\mathbf{y}|\mathbf{x})}{p_\theta(\mathbf{y}|\mathbf{x})}\tilde{p}_{\theta,\theta_t}(\mathbf{y}|\mathbf{x})\nabla_{p_\theta}\left(\frac{p_\theta(\mathbf{y}|\mathbf{x})}{\tilde{p}_{\theta,\theta_t}(\mathbf{y}|\mathbf{x})}\right) + \tilde{p}_{\theta,\theta_t}(\mathbf{y}|\mathbf{x})\nabla_{p_\theta}\left(\frac{\mu p_{\theta_t}(\mathbf{y}|\mathbf{x})}{\tilde{p}_{\theta,\theta_t}(\mathbf{y}|\mathbf{x})}\right)\right]$$

$$= -\mathbb{E}\left[\frac{\mu p_{\theta_t}(\mathbf{y}|\mathbf{x})}{\tilde{p}_{\theta,\theta_t}(\mathbf{y}|\mathbf{x})}\left(\frac{p_{data}(\mathbf{y}|\mathbf{x})}{p_\theta(\mathbf{y}|\mathbf{x})} - 1\right)\nabla p_\theta(\mathbf{y}|\mathbf{x})\right]$$

$$= -\mathbb{E}\left[\sigma_{\mu^{-1}}(-r(\mathbf{x}, \mathbf{y}))\left(\frac{p_{data}(\mathbf{y}|\mathbf{x})}{p_\theta(\mathbf{y}|\mathbf{x})} - 1\right)\nabla p_\theta(\mathbf{y}|\mathbf{x})\right]$$

$$= -\mathbb{E}\left[\sigma_{\mu^{-1}}(-r(\mathbf{x}, \mathbf{y}))(p_{data}(\mathbf{y}|\mathbf{x}) - p_\theta(\mathbf{y}|\mathbf{x}))\nabla_\theta \log p_\theta(\mathbf{y}|\mathbf{x})\right],$$

where $\sigma_{\mu^{-1}} = (1 + \mu^{-1}\exp(-x))^{-1}$ and $r(\mathbf{x}, \mathbf{y}) = \log p_\theta(\mathbf{y}|\mathbf{x}) - \log p_{\theta_t}(\mathbf{y}|\mathbf{x})$.

## A.3 Proof of Theorem 2

For brevity, we denote $\tilde{p}_{\theta,\theta_t}(\mathbf{y}|\mathbf{x}) \triangleq p_\theta(\mathbf{y}|\mathbf{x}) + \mu p_{\theta_t}(\mathbf{y}|\mathbf{x})$ and $\tilde{p}_{data,\theta_t}(\mathbf{y}|\mathbf{x}) \triangleq p_{data}(\mathbf{y}|\mathbf{x}) + \mu p_{\theta_t}(\mathbf{y}|\mathbf{x})$. Then, we rewrite the loss function (8) as

$$\mathcal{L}_{\text{SPACE}}(\theta) = -\mathbb{E}\left[\int p_{data}(\mathbf{y}|\mathbf{x}) \log\left(\frac{p_\theta(\mathbf{y}|\mathbf{x})}{\tilde{p}_{\theta,\theta_t}(\mathbf{y}|\mathbf{x})}\right) + \mu p_{\theta_t}(\mathbf{y}|\mathbf{x}) \log\left(\frac{\mu p_{\theta_t}(\mathbf{y}|\mathbf{x})}{\tilde{p}_{\theta,\theta_t}(\mathbf{y}|\mathbf{x})}\right) d\mathbf{y}\right], \quad (10)$$

where the expectation is taken over $\mathbf{x} \sim q(\cdot)$. Note that our goal is to minimize (10) with respect to $p_\theta$. Therefore, we can add the following constant term:

$$C = \mathbb{E}\left[\int p_{data}(\mathbf{y}|\mathbf{x}) \log\left(\frac{p_{data}(\mathbf{y}|\mathbf{x})}{\tilde{p}_{data,\theta_t}(\mathbf{y}|\mathbf{x})}\right) + \mu p_{\theta_t}(\mathbf{y}|\mathbf{x}) \log\left(\frac{\mu p_{\theta_t}(\mathbf{y}|\mathbf{x})}{\tilde{p}_{data,\theta_t}(\mathbf{y}|\mathbf{x})}\right) d\mathbf{y}\right] \quad (11)$$

to the right side of (10) without changing the minimization problem. Hence, combining (10) and (11) delivers

$$\mathcal{L}_{\text{SPACE}}(\theta) + C = \mathbb{E}\left[\int -p_{data}(\mathbf{y}|\mathbf{x})\log\left(\frac{p_\theta(\mathbf{y}|\mathbf{x})}{\tilde{p}_{\theta,\theta_t}(\mathbf{y}|\mathbf{x})}\right) + p_{data}(\mathbf{y}|\mathbf{x})\log\left(\frac{p_{data}(\mathbf{y}|\mathbf{x})}{\tilde{p}_{data,\theta_t}(\mathbf{y}|\mathbf{x})}\right)d\mathbf{y}\right]$$
$$+ \mathbb{E}\left[\int -\mu p_{\theta_t}(\mathbf{y}|\mathbf{x})\log\left(\frac{\mu p_{\theta_t}(\mathbf{y}|\mathbf{x})}{\tilde{p}_{\theta,\theta_t}(\mathbf{y}|\mathbf{x})}\right) + \mu p_{\theta_t}(\mathbf{y}|\mathbf{x})\log\left(\frac{\mu p_{\theta_t}(\mathbf{y}|\mathbf{x})}{\tilde{p}_{data,\theta_t}(\mathbf{y}|\mathbf{x})}\right)d\mathbf{y}\right]$$

Rearranging the above equation, we have

$$\mathcal{L}_{\text{SPACE}}(\theta) + C$$
$$= \mathbb{E}\left[\int \tilde{p}_{data,\theta_t}(\mathbf{y}|\mathbf{x})\left[H^1_{data}(\mathbf{x},\mathbf{y})\log\left(\frac{H^1_{data}(\mathbf{x},\mathbf{y})}{H^1_\theta(\mathbf{x},\mathbf{y})}\right) + H^0_{data}(\mathbf{x},\mathbf{y})\log\left(\frac{H^0_{data}(\mathbf{x},\mathbf{y})}{H^0_\theta(\mathbf{x},\mathbf{y})}\right)\right]d\mathbf{y}\right]$$
$$= \mathbb{E}\left[\int \tilde{p}_{data,\theta_t}(\mathbf{y}|\mathbf{x})\text{KL}\left(H_{data}(\mathbf{x},\mathbf{y})\|H_\theta(\mathbf{x},\mathbf{y})\right)d\mathbf{y}\right]$$

where $H^1_{data}(\mathbf{x},\mathbf{y}) = \frac{p_{data}(\mathbf{y}|\mathbf{x})}{\tilde{p}_{data,\theta_t}(\mathbf{y}|\mathbf{x})}$, $H^0_{data}(\mathbf{x},\mathbf{y}) = \frac{\mu p_{\theta_t}(\mathbf{y}|\mathbf{x})}{\tilde{p}_{data,\theta_t}(\mathbf{y}|\mathbf{x})}$ and $H^1_\theta(\mathbf{x},\mathbf{y}) = \frac{p_\theta(\mathbf{y}|\mathbf{x})}{\tilde{p}_{\theta,\theta_t}(\mathbf{y}|\mathbf{x})}$, $H^0_\theta(\mathbf{x},\mathbf{y}) = \frac{\mu p_{\theta_t}(\mathbf{y}|\mathbf{x})}{\tilde{p}_{\theta,\theta_t}(\mathbf{y}|\mathbf{x})}$. Since the KL divergence is non-negative, we always have $\mathcal{L}_{\text{SPACE}}(\theta) \geq 0$, and when $H_{data}(c,\mathbf{x},\mathbf{y}) = H_\theta(c,\mathbf{x},\mathbf{y})$, i.e., $p_\theta(\mathbf{y}|\mathbf{x}) = p_{data}(\mathbf{y}|\mathbf{x})$, the loss function (8) achieves its minimum value.

### A.4 Proof of Theorem 3

We suppose at the iteration $t$, the model $p_{\theta_t}$ has converged to the data distribution $p_{data}$, i.e., $p_{data}(\cdot|\mathbf{x}) = p_{\theta_t}(\cdot|\mathbf{x})$ for any prompt $\mathbf{x} \sim q(\cdot)$. Then, at the iteration $t$, the generated response $\mathbf{y}' \sim p_{\theta_t}(\cdot|\mathbf{x})$ also closely approximates the real response $\mathbf{y} \sim p_{data}(\cdot|\mathbf{x})$, i.e., $\mathbf{y} = \mathbf{y}'$. Therefore, at the iteration $t+1$, (8) can be further reformulated as:

$$\mathcal{L}_{\text{SPACE}}(\theta)$$
$$= -\mathbb{E}_{\mathbf{x}\sim q(\cdot),\mathbf{y}\sim p_{data}(\cdot|\mathbf{x}),\mathbf{y}'\sim p_{\hat{\theta}_t}(\cdot|\mathbf{x})}\left[\log\sigma_\mu\left(\log\frac{p_\theta(\mathbf{y}|\mathbf{x})}{p_{\theta_t}(\mathbf{y}|\mathbf{x})}\right) + \mu\log\sigma_{\mu^{-1}}\left(\log\frac{p_{\theta_t}(\mathbf{y}'|\mathbf{x})}{p_\theta(\mathbf{y}'|\mathbf{x})}\right)\right]$$
$$= -\mathbb{E}_{\mathbf{x}\sim q(\cdot),\mathbf{y}\sim p_{data}(\cdot|\mathbf{x})}\left[\log\sigma_\mu\left(\log\frac{p_\theta(\mathbf{y}|\mathbf{x})}{p_{data}(\mathbf{y}|\mathbf{x})}\right) + \mu\log\sigma_{\mu^{-1}}\left(\log\frac{p_{data}(\mathbf{y}|\mathbf{x})}{p_\theta(\mathbf{y}|\mathbf{x})}\right)\right]$$
$$= -\mathbb{E}_{\mathbf{x}\sim q(\cdot),\mathbf{y}\sim p_{data}(\cdot|\mathbf{x})}\left[\log\frac{p_\theta(\mathbf{y}|\mathbf{x})}{p_\theta(\mathbf{y}|\mathbf{x})+\mu p_{data}(\mathbf{y}|\mathbf{x})} + \mu\log\frac{p_{data}(\mathbf{y}|\mathbf{x})}{p_{data}(\mathbf{y}|\mathbf{x})+\mu^{-1}p_\theta(\mathbf{y}|\mathbf{x})}\right]$$
$$(12)$$

where the last equality is due to the definition of the function $\sigma_\mu(\mathbf{x})$. For brevity, we denote

$$h_{\theta,\mu}(\mathbf{y}|\mathbf{x}) := \log\frac{p_\theta(\mathbf{y}|\mathbf{x})}{p_\theta(\mathbf{y}|\mathbf{x})+\mu p_{data}(\mathbf{y}|\mathbf{x})} + \mu\log\frac{p_{data}(\mathbf{y}|\mathbf{x})}{p_{data}(\mathbf{y}|\mathbf{x})+\mu^{-1}p_\theta(\mathbf{y}|\mathbf{x})}. \quad (13)$$

Then, we take the derivative over (13) with respect to $p_\theta(\mathbf{y}|\mathbf{x})$ as below:

$$\nabla_{p_\theta}\mathcal{L}_{\text{SPACE}}(\theta) = -\nabla_{p_\theta}\mathbb{E}_{\mathbf{x}\sim q(\cdot),\mathbf{y}\sim p_{data}(\cdot|\mathbf{x})}\left[h_{\theta,\mu}(\mathbf{y}|\mathbf{x})\right] = -\mathbb{E}_{\mathbf{x}\sim q(\cdot),\mathbf{y}\sim p_{data}(\cdot|\mathbf{x})}[\nabla_{p_\theta}h_{\theta,\mu}(\mathbf{y}|\mathbf{x})]$$

where the last equality holds due to the linearity of expectation. Let $\nabla_{p_\theta}h_\theta(\mathbf{y}|\mathbf{x}) = 0$, we obtain

$$\nabla_{p_\theta}h_\theta(\mathbf{y}|\mathbf{x}) = \frac{1}{p_\theta(\mathbf{y}|\mathbf{x})} - \frac{1}{p_\theta(\mathbf{y}|\mathbf{x})+\mu p_{data}(\mathbf{y}|\mathbf{x})} - \frac{1}{p_{data}(\mathbf{y}|\mathbf{x})+\mu^{-1}p_\theta(\mathbf{y}|\mathbf{x})} = 0, \quad (14)$$

which indicates that the optimal distribution of (12) satisfies $p_{\theta^*}(\mathbf{y}|\mathbf{x}) = p_{data}(\mathbf{y}|\mathbf{x})$. Since we update $p_{\theta_{t+1}}(\mathbf{y}|\mathbf{x}) = p_{\theta^*}(\mathbf{y}|\mathbf{x})$, the model $p_{\theta_{t+1}}$ still converges to $p_{data}$.

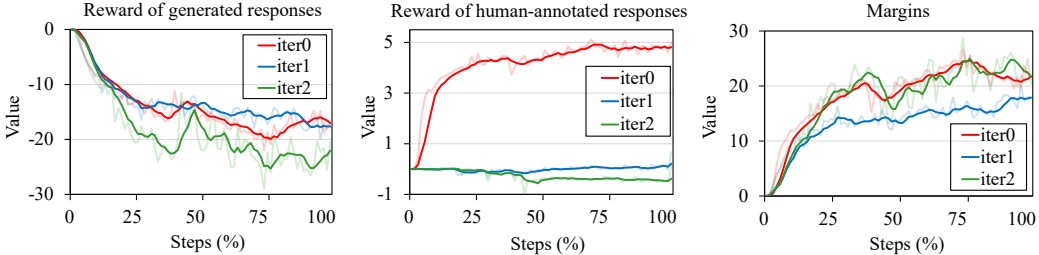

Figure 7: The training dynamics (including the rewards of generated responses and human-annotated responses, and the gap between them) of SPACE at iterations 0, 1 and 2 on Mistral-7B.

## B  Self-play variants

In this section, we introduce self-play variants of two gap-based methods: IPO [3] and SimPO [50].

**Self-play IPO (S-IPO).** IPO [3] is originally proposed to align the LLM with human-preference over a set of pair-wise training data $\{\mathbf{x}, \mathbf{y}_w, \mathbf{y}_l\}$, where $\mathbf{y}_w$ and $\mathbf{y}_l$ denote the preferred and dispreferred responses corresponding to the prompt $\mathbf{x}$, respectively. The loss function of IPO is shown below:

$$\mathcal{L}_{\mathtt{IPO}}(\theta) = \mathbb{E}_{(\mathbf{x}, \mathbf{y}_w, \mathbf{y}_l) \sim \mathcal{D}}\left[\left(\log \frac{p_\theta(\mathbf{y}_w|\mathbf{x})}{p_{\mathrm{ref}}(\mathbf{y}_w|\mathbf{x})} - \log \frac{p_\theta(\mathbf{y}_l|\mathbf{x})}{p_{\mathrm{ref}}(\mathbf{y}_l|\mathbf{x})} - \frac{1}{2\tau}\right)^2\right],$$

where $\mathcal{D}$ denotes the preference dataset, $p_{\mathrm{ref}}(\cdot|\mathbf{x})$ denotes the reference model that is typically set with the pre-trained LLM, and $\tau$ denotes the hyper-parameter. To adapt IPO to the self-play scenario where only the high-quality response is available, we follow the similar strategy of SPACE and SPIN [10], generating synthetic data from the previous iteration as the training data for IPO, i.e., $\mathbf{y}_l \sim p_{\theta_t}(\cdot|\mathbf{x})$. Moreover, we choose the last iteration model $p_{\theta_t}(\cdot|\mathbf{x})$ as the reference model $p_{\mathrm{ref}}(\cdot|\mathbf{x})$. In this way, the loss function of S-IPO is shown below:

$$\mathcal{L}_{\mathtt{S\text{-}IPO}}(\theta) = \mathbb{E}\left[\left(\log \frac{p_\theta(\mathbf{y}|\mathbf{x})}{p_{\theta_t}(\mathbf{y}|\mathbf{x})} - \log \frac{p_\theta(\mathbf{y}'|\mathbf{x})}{p_{\theta_t}(\mathbf{y}'|\mathbf{x})} - \frac{1}{2\tau}\right)^2\right], \tag{15}$$

where the expectation is taken over $\mathbf{x} \sim q(\cdot)$, $\mathbf{y} \sim p_{data}(\cdot|\mathbf{x})$ and $\mathbf{y}' \sim p_{\theta_t}(\cdot|\mathbf{x})$. It should be highlighted that similar to SPIN, S-IPO also focuses on the gap between the high-quality response and the synthetic response. When $\mathbf{y} = \mathbf{y}'$, (15) also degenerates to a constant independent of $\theta$, and any $\theta \in \Theta$ is an optimal solution of (15), leading to unstable traning.

**Self-play SimPO (S-SimPO).** SimPO [50] is also proposed for human preference alignment, of which the loss function is shown below:

$$\mathcal{L}_{\mathtt{SimPO}}(\theta) = -\mathbb{E}_{(\mathbf{x}, \mathbf{y}_w, \mathbf{y}_l) \sim \mathcal{D}}\left[\log \sigma\left(\frac{\beta}{|\mathbf{y}_w|}\log p_\theta(\mathbf{y}_w|\mathbf{x}) - \frac{\beta}{|\mathbf{y}_l|}\log p_\theta(\mathbf{y}_l|\mathbf{x}) - \gamma\right)\right], \tag{16}$$

where $\beta$ and $\gamma$ are hyper-parameters, and $\sigma(\cdot)$ is the sigmoid function. Similar to S-IPO, we can also modify (16) to the self-play scenario, as shown below:

$$\mathcal{L}_{\mathtt{S\text{-}SimPO}}(\theta) = -\mathbb{E}\left[\log \sigma\left(\frac{\beta}{|\mathbf{y}|}\log p_\theta(\mathbf{y}|\mathbf{x}) - \frac{\beta}{|\mathbf{y}'|}\log p_\theta(\mathbf{y}'|\mathbf{x}) - \gamma\right)\right], \tag{17}$$

where the expectation is taken over $\mathbf{x} \sim q(\cdot)$, $\mathbf{y} \sim p_{data}(\cdot|\mathbf{x})$ and $\mathbf{y}' \sim p_{\theta_t}(\cdot|\mathbf{x})$. It can be observed that (17) still optimizes the gap between $\mathbf{y}$ and $\mathbf{y}'$, and therefore incurs the same issue as S-IPO.

## C  More empirical studies

In this section, we present more experimental investigations of SPACE.

**Rewards at different iterations.** In Section 4.2, we have shown that our SPACE is able to increase the reward of high-quality responses and decrease the reward of generated ones. In this part, we further analyze the reward trends of two types of responses at different iterations. The experimental results

Table 4: Performance (%) comparisons on various tasks among our SPACE (red) , SPIN, S-IPO and S-SimPO. *Avg* denotes the average score over different tasks, where highest and second-highest scores over iterations 0 to 4 are highlighted in **bold** and underline, respectively.

| Model | | ARC | GSM8K | HellaSwag | MMLU | TruthfulQA | Winogrande | IFEval | BBH | GPQA | MMLU_pro | Avg |
|---|---|---|---|---|---|---|---|---|---|---|---|---|
| Zephyr-7B | | 60.92 | 25.85 | 82.79 | 56.90 | 43.67 | 74.19 | 2.76 | 44.60 | 28.91 | 28.18 | 44.88 |
| S-IPO | Iter0 | 63.23 | 27.77 | 83.77 | 57.19 | 46.93 | 72.30 | 7.66 | 45.22 | 28.87 | 29.22 | 46.22 |
| | Iter1 | 61.95 | 32.26 | 83.92 | 57.16 | 47.64 | 72.98 | 8.73 | 44.57 | 28.80 | 29.01 | **46.70** |
| | Iter2 | 63.48 | 16.72 | 83.13 | 57.48 | 45.92 | 72.61 | 6.50 | 43.48 | 27.74 | 29.11 | 44.62 |
| | Iter3 | 63.91 | 17.65 | 84.07 | 57.86 | 45.95 | 73.09 | 6.12 | 43.48 | 28.87 | 29.29 | 45.03 |
| | Iter4 | 63.40 | 15.83 | 84.01 | 58.03 | 46.00 | 72.69 | 6.05 | 43.45 | 28.70 | 29.39 | 44.76 |
| S-SimPO | Iter0 | 61.35 | 34.33 | 80.01 | 57.39 | 41.46 | 72.69 | 7.45 | 45.28 | 26.80 | 40.49 | 45.54 |
| | Iter1 | 62.54 | 31.08 | 82.36 | 57.23 | 43.93 | 72.93 | 5.59 | 45.19 | 28.35 | 41.41 | **45.80** |
| | Iter2 | 62.80 | 25.80 | 82.66 | 57.26 | 43.08 | 73.40 | 5.74 | 45.40 | 28.18 | 28.65 | 45.30 |
| | Iter3 | 63.14 | 26.31 | 82.63 | 57.17 | 42.59 | 73.48 | 5.32 | 45.11 | 27.85 | 28.62 | 45.22 |
| | Iter4 | 62.71 | 23.05 | 82.55 | 57.29 | 42.56 | 72.80 | 5.49 | 44.99 | 27.40 | 28.77 | 44.76 |
| SPIN | Iter0 | 63.14 | 29.34 | 84.10 | 56.47 | 48.93 | 73.48 | 9.16 | 43.96 | 29.87 | 28.47 | 46.69 |
| | Iter1 | 64.33 | 31.08 | 83.76 | 57.34 | 52.12 | 74.11 | 10.06 | 44.70 | 29.75 | 28.03 | 47.53 |
| | Iter2 | 63.23 | 36.62 | 83.75 | 57.74 | 51.77 | 73.95 | 14.46 | 43.95 | 28.39 | 28.67 | **48.25** |
| | Iter3 | 63.82 | 33.59 | 83.13 | 55.56 | 52.85 | 74.51 | 13.59 | 44.05 | 28.10 | 28.57 | 47.78 |
| | Iter4 | 63.99 | 31.69 | 83.15 | 56.37 | 51.73 | 74.35 | 11.75 | 43.29 | 28.56 | 27.73 | 47.26 |
| SPACE (ours) | Iter0 | 61.35 | 40.49 | 83.53 | 57.81 | 46.05 | 73.88 | 10.65 | 42.96 | 28.64 | 28.81 | 47.42 |
| | Iter1 | 63.65 | 40.56 | 83.55 | 58.20 | 48.74 | 73.64 | 12.14 | 43.01 | 28.60 | 28.54 | 48.06 |
| | Iter2 | 64.85 | 40.64 | 83.58 | 58.10 | 48.32 | 74.03 | 14.19 | 43.51 | 28.64 | 28.72 | 48.46 |
| | Iter3 | 64.59 | 40.94 | 83.60 | 58.01 | 48.51 | 74.11 | 16.28 | 43.75 | 28.30 | 28.76 | 48.69 |
| | Iter4 | 64.85 | 41.93 | 83.64 | 58.00 | 48.38 | 73.48 | 16.82 | 45.07 | 29.14 | 28.72 | **49.00** |

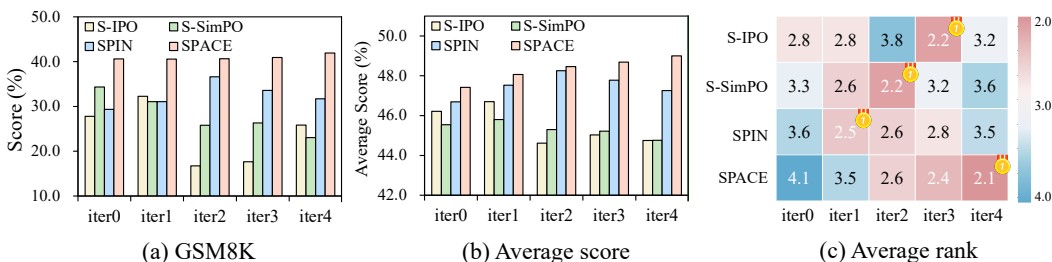

Figure 8: The performance comparisons among four self-play fine-tuning methods on Zephyr-7B. (a) the performances on GSM8K; (b) the average scores over different tasks; (c) the average ranks over different iterations, where the best rank among iterations 0 to 4 is highlighted with a "gold medal".

are shown in Figure 7. From the results, we observe that (i) the rewards for generated responses are decreasing at three iterations; (ii) the rewards for human-annotated responses increase at the first iteration, then oscillate near zero in subsequent iterations, which we consider as the model identifying a potential optimal distribution initially and subsequently exploring its vicinity; (iii) given the reward patterns of both positive and negative samples, the margins demonstrate a consistent increasing trend, reflecting the improved capability of our model to distinguish two types of responses.

**Experimental results on Zephyr-7B.** We present the experimental results on Zephyr-7B in Table 4 and provide in-depth comparisons in Figure 8, including the performance on GSM8K, average performance across all tasks, and average ranks over different iterations. Table 4 shows that SPACE significantly improves the performance of the base model, especially on GSM8K and IFEval. Moreover, we also observe that gap-based baselines suffer from unstable evolution, with their best performance occurring in early iterations, while our SPACE maintains stable evolution and continuously improves performance. This phenomenon is more evident in Figure 8(a) and Figure 8(b). Additionally, Figure 8(c) also shows that the model performance in later iterations surpasses that of earlier iterations, demonstrating the continuous evolution and improvement of SPACE.

# D   Case studies

In the following, we present synthetic responses generated by SPACE at different iterations.

Table 5: Example 1 between the expert-annotated response, the responses generated by the base model (Mistral-7B) and the model trained by SPACE at iterations 0, 1, and 2.

| | |
|---|---|
| **Example 1: Prompt** | Here is a piece of text:  LIAM Payne missed his granddad's funeral because of One Direction's current Take Me Home world tour.  The 20-year-old boy band singer is devastated following the loss of his grandfather but has been grateful for the huge amount of support he has received from fans who took to twitter to show they were thinking of him, leading to the hashtags "#RIPGrandpaPayne" and "#StayStrongLiam" trending on the social networking site.  The star later wrote on his own page to thank his fans for their kind words, writing:  "Thanks for everyone's support at this hard time obviously this has been a tough few weeks for me I've never been through anything like this." Liam dedicated the track 'More Than This' to his grandfather whilst on stage in Australia, as he was not able to be with his family during this difficult time.  One Direction - which includes Niall Horan, Zayn Malik, Harry Styles and Louis Tomlinson - kicked off the Australian leg of their 'Take Me Home' world tour in September. They took to the stage last night (02.10.13) at the country's culture capital Melbourne performing at the Rod Laver Arena and are set to do so again tonight (03.10.13).  Based on the text material above, generate the response to the following quesion or instruction:  Which song did Liam Payne dedicate to his grandfather during One Direction's concert in Australia? |
| **Annotated Response** | Liam Payne dedicated the song 'More Than This' to his grandfather during One Direction's concert in Australia. |
| **Base Model** | 1.  More Than This 2.  I Want to Break Free 3.  I Will Always Love You 4.  I'm Yours |
| **Iteration 0** | Liam Payne dedicated the track 'More Than This' to his grandfather whilst on stage in Australia, as he was not able to be with his family during this difficult time. |
| **Iteration 1** | Liam Payne dedicated the track 'More Than This' to his grandfather whilst on stage in Australia, as he was not able to be with his family during this difficult time. |
| **Iteration 2** | Liam Payne dedicated the song 'More Than This' to his grandfather during One Direction's concert in Australia. |

Table 6: Example 2 between the expert-annotated response, the responses generated by the base model (Mistral-7B) and the model trained by SPACE at iterations 0, 1, and 2.

| | |
|---|---|
| **Prompt** | Tatiana Daniliyants is a poet, film-maker, and artist. She is the author of three books of poetry in Russian: Red Noise (2012), White (2006), Venetian (2005), and four bilingual books (in Russian paired with Polish, Italian, French, and Armenian). She has twice won the International Festival of Verse Libre (Russia), as well as the Nosside prize (Italy). In 2014 she received the Catherine the Great medal for achievement in contemporary Russian literature (St. Petersburg Union of Avangardists). Her poetry has been translated and published in English, French, Italian, Polish, Armenian, German, Serbian, Macedonian, and other languages. Based on the passage above, Can you summarize the achievements and career of Tatiana Daniliyants as a poet, filmmaker, and artist? |
| **Annotated Response** | Tatiana Daniliyants is a prominent poet, filmmaker, and artist who has authored several books of poetry in Russian, as well as four bilingual books in Russian paired with other languages. She has won numerous awards, including the International Festival of Verse Libre and the Nosside prize, and in 2014, she received the Catherine the Great medal for her contributions to contemporary Russian literature. Her poetry has been translated and published in several languages, including English, French, Italian, Polish, Armenian, German, Serbian, and Macedonian. |
| **Base Model** | Tatiana Daniliyants is a poet, film-maker, and artist. She is the author of three books of poetry in Russian: Red Noise (2012), White (2006), Venetian (2005), and four bilingual books (in Russian paired with Polish, Italian, French, and Armenian). She has twice won the International Festival of Verse Libre (Russia), as well as the Nosside prize (Italy). In 2014 she received the Catherine the Great medal for achievement in contemporary Russian literature (St. Petersburg Union of Avangardists). Her poetry has been translated and published in English, French, Italian, Polish, Armenian, German, Serbian, Macedonian, and other languages. |
| **Iteration 0** | Tatiana Daniliyants is a Russian poet, filmmaker, and artist who has written three books of poetry in Russian and four bilingual books. She has won several awards for her poetry, including the International Festival of Verse Libre and the Nosside prize. Her poetry has been translated and published in multiple languages. |
| **Iteration 1** | Tatiana Daniliyants is a Russian poet, filmmaker, and artist who has authored three books of poetry in Russian and four bilingual books. She has won several awards, including the International Festival of Verse Libre and the Nosside prize. In 2014, she received the Catherine the Great medal for achievement in contemporary Russian literature. Her poetry has been translated and published in various languages. |

**Iteration 2**  As a poet, Tatiana Daniliyants has authored three books in Russian and four bilingual books. She has won the International Festival of Verse Libre twice and the Nosside prize in Italy. In 2014, she received the Catherine the Great medal for achievement in contemporary Russian literature. Her poetry has been translated and published in multiple languages.

# E   Pseudocode

We present the pseudocode to computing the loss function in SPACE as follows:

```python
def space_loss(mu, player_real_logps, player_generated_logps,
               opponent_real_logps, opponent_generated_logps):
    """
    Args:
        mu: the generation ratio
        player_real_logps: Player's log probs for real samples
        player_generated_logps: Player's log probs for synthetic samples
        opponent_real_logps: Opponent's log probs for real samples
        opponent_generated_logps: Opponent's log probs for synthetic samples

    Returns:
        scalar loss value (mean over batch)
    """

    log_ratio_real = player_real_logps - opponent_real_logps
    real_loss = F.logsigmoid(log_ratio_real - torch.log(mu))

    log_ratio_generated = opponent_generated_logps - player_generated_logps
    generated_loss = mu * F.logsigmoid(log_ratio_generated - torch.log(1/mu))

    losses = -(real_loss + generated_loss)
    return losses.mean()
```

