# OpenReview forum: "SPACE: Noise Contrastive Estimation Stabilizes Self-Play Fine-Tuning for Large Language Models"
_NeurIPS.cc/2025/Conference — NeurIPS 2025 poster_

### Official Review · Reviewer_j1nk · 2025-07-01

**Clarity:** 2
**Significance:** 2
**Originality:** 2
**Rating:** 3
**Confidence:** 4

**Summary:**

This paper proposes the noise contrastive estimation-based self-play fine-tuning approach, called SPACE. This method addresses the training instability issue of SPIN, a prior method in this direction. SPACE independently optimizes the absolute reward values for each type of data, ensuring a consistently meaningful objective and thereby avoiding the instability issue. Empirical results on Mistral and Zephyr demonstrate the advantages of the proposed method.

**Questions:**

1. Given the both SFT, SPIN, SPACE can theoretically converge to the underlying data distribution, do you believe what are exactly advantages of self-playing algorithms in this case?
2. Do you believe the performance degradation in Figure 2 for SPIN is due to optimization instability or generalization issues?
3. Have you tested the method on Qwen2.5 and Llama-3.1 series? Models like Zephyr and Mistral are outdated (e.g., scoring 40-50 on simple benchmarks like GSM8K), making it hard to justify the superiority of the proposed method.
4. The OpenLLM Leaderboard primarily evaluates few-shot in-context learning (ICL) capabilities rather than downstream performance. Have you tested more challenging tasks like MATH?
5. Have you tested other datasets to ensure broader applicability of your conclusions?
6. A limitation of self-play algorithms is overfitting to the empirical data distribution, reducing diversity and harming subsequent RL fine-tuning. Have the authors observed this issue, where model outputs heavily conform to empirical data patterns?

**Ethical Concerns:**

["NO or VERY MINOR ethics concerns only"]

**Final Justification:**

While the authors have provided additional empirical evidence, these results unfortunately demonstrate only marginal improvements, thereby calling into question the overall effectiveness of their proposed methods. Consequently, I maintain my original recommendation regarding this work.

**Limitations:**

Yes

**Quality:**

2

**Strengths And Weaknesses:**

Strengths:
- The paper is well-written in most places and easy to follow.
- The provided empirical results seem promising.

Weakness:
- The criticism of SPIN seems only empirical, which lacks rigor. It is unclear whether this is due to a fundamental limitation or hyperparameter tuning. The argued extreme case—where the generated response matches the real one—is not a bug for me, as SPIN may already learn the data distribution in this scenario.
- Lack of related work discussion. This paper lies within the framework of statistical distribution matching but fails to discuss seminal works like Generative Adversarial Nets (GANs) and Generative Adversarial Imitation Learning (GAIL).
- Empirical results are rather weak. Mistral and Zephyr are relatively weak models, and results on these models may not generalize to modern ones.
- Empirical comparisons with Supervised Fine-Tuning (SFT) are not rigorous. The proposed method requires more compute due to synthetic data generation and additional gradient iterations. Thus, comparisons should include more epochs (even beyond 4). The results in Figure 5 do not convincingly justify the comparison with the same human-annotated data.

---

Goodfellow, Ian J., et al. "Generative adversarial nets." Advances in neural information processing systems 27 (2014).

Ho, Jonathan, and Stefano Ermon. "Generative adversarial imitation learning." Advances in neural information processing systems 29 (2016).

---

> ### Author Rebuttal · Authors · 2025-07-30
>
> **Many thanks for the constructive reviews!**
>
> ---
>
> **Q1**: The criticism (optimization instability) of SPIN seems only empirical. Do you believe the performance degradation in Figure 2 for SPIN is due to optimization instability or generalization issues?
>
> **A1**: We note that there are several studies also pointing out the training instability of SPIN [1, 2], which appears to be ***a widely-recognized issue*** in the literature. In our work, we have provided an in-depth discussion showing that gap-based methods (including SPIN) are susceptible to performance collapse when extended to self-play fine-tuning (Lines 183–195). This phenomenon is also empirically validated through extensive experiments (***Table 1***). The extreme cases presented in our analysis are intended to facilitate understanding. In practice, during the iterations, the loss of SPIN may become very small (even a constant), but the learned distribution may not align with the target distribution; instead, it could be ***any*** distribution, leading to performance collapse. We will provide a more detailed clarification in the revised version.
>
> ---
>
> **Q2**: Lack of related work discussion, such as GAN and GAIL.
>
> **A2**: Thanks for the helpful suggestion. GAN [3] is proposed to learn generative models via an adversarial process, and GAIL [4] extends this adversarial paradigm to imitation learning. Overall, both [3, 4] and SPACE adopt two-player adversarial frameworks. However, SPACE is specifically designed for self fine-tuning of LLMs, where the two players are instances of the same LLM with different parameters. We will include a detailed discussion of this connection in the revised version.
>
> ---
>
> **Q3**: Results on Mistral and Zephyr may not generalize to modern ones.
>
> **A3**: Thanks for the valuable suggestion. Here, we conduct additional empirical studies on Qwen2.5-7B-base model. The results are presented in the table below, where *base model* refers to the average performance of Qwen2.5 over multiple tasks  (the same task set as shown in Table 5; only average scores are shown due to character limitations). From the results, we observe that (i) SPIN continues to suffer from unstable iterative optimization, while SPACE demonstrates consistent performance improvements; and (ii) the performance gains achieved by SPACE on Qwen2.5 are less pronounced compared to those on Mistral and Zephyr. This may be because Qwen2.5 has already been well optimized during pretraining, leaving less room for further improvement when only 50K annotated examples are used.
>
> ||base model|iter0|iter1|iter2|iter3|iter4|
> | -------| ------------| -------| -------| -------| -------| -------|
> |SPIN|59.18|59.61|59.81|59.63|59.69|59.89|
> |SPACE|59.18|60.06|60.46|60.53|60.54|60.54|
>
> ---
>
> **Q4**: Empirical comparisons with SFT are not rigorous.
>
> **A4**: Thanks for the thoughtful consideration. Figure 5 is intended to highlight that SPACE, using a small amount of annotated data, outperforms SFT trained on a significantly larger number of annotated examples. In Figure 5, we present the performance of SPACE trained ***over two iterations***. For example, the reported average performance 51.48 is measure on the model trained with a total of 150K samples, including 50K annotated data and 100K synthetic data. To ensure a fair comparison under the same computational budget, we run SFT for multiple rounds with more annotated samples, and present the results in the following table. The results show that, even when using more annotated data, SFT still underperforms compared to SPACE (51.48 with a total of 150K samples).
>
> ||400K|600K|800K|
> | -----| -------| -------| -------|
> |SFT|50.48|50.14|49.84|
>
> ---
>
> **Q5**: Advantages of self-playing algorithms.
>
> **A5**: We appreciate your valuable question! We summarize the advantages of self-play methods and SPACE as follows:
>
> * **Better Performance with Fewer Annotations.**  SFT typically requires a large amount of annotated data, which is often impractical in real-world scenarios, and thus motivates the development of self-play fine-tuning methods. In this paper, we theoretically prove that with limited annotated samples, SPACE is able to capture the target data distribution (***Theorem 2***). Moreover, empirical results demonstrate that SPACE achieves superior performance with a small amount of annotated data, compared to SFT trained on significantly larger datasets (***Figure 4***).
> * **Stable Performance Improvement.**  Compared to the existing self-play method SPIN, the primary advantage of SPACE lies in its stable iterative optimization. To support this, we first provide a theoretical analysis demonstrating that SPACE is able to maintain the optimal distribution (***Theorem 3***). We then conduct extensive empirical studies to validate the stable convergence of SPACE over iterations (***Table 1***).
>
>
> Additionally, we would like to emphasize that while SFT, SPIN, and SPACE can theoretically capture the underlying data distribution, they differ in their convergence rates (i.e., the amount of annotated data required) and stability during optimization. We will include a detailed discussion in the revised version.
>
> ---
>
> **Q6**: Test more challenging tasks like MATH and other datasets.
>
> **A6**: Thanks for the helpful suggestion! We have conducted additional experiments on MATH [5], openbookqa [6], and MT-Bench [7], which evaluate mathematical reasoning, common knowledge, and conversational ability in open-ended scenarios, respectively. We present the results in the following table, where columns 3 to 7 correspond to iterations 0 to 4 of SPACE. From the table, we observe that SPACE can progressively improve performance while maintaining stability.
>
> ||Mistral|iter0|iter1|iter2|iter3|iter4|
> | ------------| ---------| -------| -------| -------| -------| -------|
> |MATH|3.25|3.40|3.58|3.69|3.84|3.90|
> |openbookqa|44.0|44.4|45.0|45.4|45.4|46.0|
> |MT-Bench|3.56|5.87|6.13|6.27|6.43|6.44|
>
> ---
>
> **Q7**: Have the authors observed the overfitting of self-play methods?
>
> **A7**: Thanks for the insightful question. We acknowledge that this concern is well-founded, since self-play methods are designed to handle the scenarios with scarce annotated data, where fine-tuning LLMs while maintaining diversity over multiple benchmarks is inherently challenging.  In our experiments, we observe that SPACE indeed demonstrates relatively marginal performance improvements on certain benchmarks, such as Winogrande. However, it is important to note that this is ***a common issue*** with self-play methods, not just specific to our SPACE. Nevertheless, we would like to emphasize that SPACE is proposed to address the instability issue of SPIN, and we consider the investigation of diversity for self-play methods a meaningful direction and leave it as the future work.
>
> ---
>
> **Reference.**
>
> [1] Investigating regularization of self-play language models. 2024.
>
> [2] Dynamic noise preference optimization for LLM self-Improvement via synthetic data. 2025.
>
> [3] Generative adversarial nets. 2014.
>
> [4] Generative adversarial imitation learning. 2016.
>
> [5] Measuring mathematical problem solving with the math dataset. 2021.
>
> [6] Can a suit of armor conduct electricity? a new dataset for open book question answering. 2018.
>
> [7] Judging llm-as-a-judge with mt-bench and chatbot arena. 2023.
>
> ---
>
> We will carefully revise the paper according to your suggestions. If our responses have properly addressed your concerns, we would appreciate it so much if you could ***re-evaluate*** our work. We are also happy to provide further clarifications during the reviewer-author discussions if needed.

---

> > ### Comment · Reviewer_j1nk · 2025-08-05
> >
> > Thank you for the clarification and the provided results. However, I still have concerns about the method's effectiveness.
> >
> > For example, the empirical improvement on Qwen2.5-7B appears limited despite leveraging more synthetic data. Typically, incorporating online rollouts with RL should yield more significant gains. This might be due to evaluation on few-shot learning tasks, but I’m uncertain. This raises doubts about the approach's effectiveness beyond older models like Mistral.
> >
> > Additionally, it is surprising that SFT with more data does not improve performance and may even degrade it. Is there an explanation for this behavior?

---

> > > ### Author Response · Authors · 2025-08-06
> > >
> > > **Many thanks for the follow-up questions!**
> > >
> > > ---
> > >
> > > **Q1**: Improvements on Qwen2.5-7B.
> > >
> > > **A1**: Thanks for the thoughtful question. We present our responses as follows:
> > >
> > > * First, we feel there are some misunderstandings.  During the training phase, both SPACE and SPIN use the same amount of data in each iteration, i.e., 50K annotated samples from Ultrachat200k and 50K synthetic samples, and ***SPACE does not use more synthetic data***.
> > >
> > > * Second, we would like to highlight the superiority of SPACE over SPIN: during 4 iterations, SPIN only improves by 0.71 points, while ***SPACE improves by 1.36 points***, which is noteworthy. Moreover, a direct comparison between our SPACE and RL methods may not be appropriate, as they are designed for different settings. We sincerely appreciate your considerations, and we plan to extend the idea of SPACE to RL research in the future.
> > >
> > > * Third, as mentioned in our rebuttal, we acknowledge that the improvements achieved by SPACE on Qwen2.5-7B are relatively small compared to Mistral. This may be due to: (i) during the pre-training phase, ***Qwen2.5 has encountered a large amount of data that are potentially similar to those in benchmarks***, leaving limited room for further improvement; (ii) we focus on the scenarios with limited annotated data, where ***achieving significant performance enhancements is inherently challenging***. In the experiments, to ensure consistency, we follow the setup in SPIN, using 50K annotated samples from Ultrachat200K as the training set.
> > >
> > > Due to limited computational resources, we are unable to provide more experimental results before the approaching discussion deadline. We plan to conduct additional experiments (including increasing the data quantity and diversity) in the revised version, to further demonstrate the effectiveness of SPACE on Qwen2.5.
> > >
> > > ---
> > >
> > > **Q2**: Performances of SFT.
> > >
> > > **A2**: We consider this degradation may be due to ***overfitting of SFT on the training data***, which could lead to a performance decline over multi-task benchmarks that evaluate the general capabilities of LLMs. Specifically, according to your suggestion, we run SFT for multiple epochs to ensure a fair comparison. Note that the training data are selected from Ultrachat200K, which contains *at most* 200K annotated samples. In the table, *800K* corresponds to running 4 epochs on 200K annotated samples. With multiple epochs of SFT on these fixed annotated data, the model may overly capture their distribution, potentially harming its generalization ability on other tasks.
> > >
> > > ---
> > >
> > > We greatly appreciate your suggestions and will incorporate them in the revised version. We hope that our responses can address your concerns and we would appreciate it so much if you could ***re-evaluate*** our work. We are also happy to provide further clarifications during the reviewer-author discussions if needed.

---

> > > > ### Comment · Reviewer_j1nk · 2025-08-07
> > > >
> > > > Thanks for clarification. I would like to make my final rating after the discussion with other reviewers and ACs.

---

> > > > > ### Author Response · Authors · 2025-08-07
> > > > >
> > > > > Many thanks for your time and effort in reviewing this paper! Please feel free to let us know if you have any more questions, and we are happy for further discussions. Your valuable suggestions will greatly improve our work.

---

### Official Review · Reviewer_Hm2z · 2025-07-03

**Clarity:** 3
**Significance:** 3
**Originality:** 3
**Rating:** 4
**Confidence:** 4

**Summary:**

Current self-play fine-tuning methods like SPIN suffer from instability during training iterations. Specifically, current methods focus on maximizing relative gaps between rewards of real and synthetic data, but when synthetic data become similar to real ones, the objective function can degrade into a meaningless constant, leading to performance collapse. Instead of comparing relative rewards, SPACE optimizes absolute reward values for real and synthetic responses separately. In this way, even when synthetic and real responses become similar, the objective remains meaningful and doesn't collapse. But there is concern about whether NCE's fundamental assumptions hold in their setting.

**Questions:**

1. What happens if synthetic data quality degrades? Especially in the early stage?
2. Could the iterative process gradually degrade capabilities in non-targeted areas?

**Ethical Concerns:**

["NO or VERY MINOR ethics concerns only"]

**Final Justification:**

During the rebuttal period, the authors handle my concern properly and I will remain my score, which is already positive.

**Limitations:**

Test with model under 7B; only one dataset; the Technical theory assumption lack of proof

**Quality:**

3

**Strengths And Weaknesses:**

Strengths:
1. Hot topic;
2. Clever Technical Insight
3. Good performance.

Weakness:
1. Is the sync data a noise? NCE's core assumption that noise remains consistently different from real data. Real noise is uninformative and random. But the sync data is in fact informative, and gets better over time. And in fact the sync data can be better than real data, especially for datasets like Ultrachat200k.
2. Limited datasets evaluated. Only Ultrachat200k is evaluated.
3. Figure 5 may be an unfair comparison on SFT and SPACE. SPACE generates more than 200K data and runs for 4 times longer than SFT. To be fair, at least SFT should be run with the same computational budget.
4. Missing recent methods. There are quite a few follow-up work after SPIN, that notices the model update issue. For example, Dynamic Noise Preference Optimization for LLM Self-Improvement via Synthetic Data https://arxiv.org/abs/2502.05400 also tries to add noise concept to resolve the model update stuck problem, although in another way.

---

> ### Author Rebuttal · Authors · 2025-07-30
>
> **Many thanks for the constructive reviews!**
>
> ---
>
> **Q1**: Is the sync data a noise?
>
> **A1**: Thanks for the insightful question. In fact, the sync data generated by the previous iteration's LLM is not ***conventional uninformative noise*** in the traditional NCE. Instead, we leverage its progressively improving nature as a self-generated supervision to guide the iterative optimization. The sync data $y'\sim p_{\theta_{t-1}}$ can be viewed as hard negatives for $p_{\theta_{t}}$, playing a critical role in the two-player framework [1, 2]. We note that, [3] demonstrates the quality of self-generated samples can potentially exceed that of annotated samples in self-play fine-tuning methods. This is also possible in SPACE. However, it is important to notice that when the sync data $y'$ aligns well with $p_{\text{data}}$, i.e., $p_{\text{data}}(y'|x) = p_{\theta_t}(y'|x)$, our SPACE can provably maintain the optimality on $y'$, as shown in ***Theorem 3***.
>
> ---
>
> **Q2**: Limited datasets evaluated.
>
> **A2**: To ensure the consistency with SPIN, we follow their experimental setup and conduct our experiments on the Ultrachat200k dataset. In addition, we have already validated the effectiveness of SPACE on many standard benchmarks, including ARC, GSM8K, and MMLU. Moreover, we have conducted additional experiments on MATH [1], openbookqa [2], and MT-Bench [3], which evaluate mathematical reasoning, common knowledge, and conversational ability in open-ended scenarios, respectively. We present the results in the following table, where columns 3 to 7 correspond to iterations 0 to 4 of SPACE. From the table, we observe that SPACE can progressively improve performance while maintaining stability.
>
> ||Mistral|iter0|iter1|iter2|iter3|iter4|
> | ------------| ---------| -------| -------| -------| -------| -------|
> |MATH|3.25|3.40|3.58|3.69|3.84|3.90|
> |openbookqa|44.0|44.4|45.0|45.4|45.4|46.0|
> |MT-Bench|3.56|5.87|6.13|6.27|6.43|6.44|
>
> ---
>
> **Q3**: Figure 5 may be an unfair comparison on SFT and SPACE.
>
> **A3**: Thanks for the insightful comment. Figure 5 is intended to highlight that SPACE, using a small amount of annotated data, outperforms SFT trained on a significantly larger number of annotated examples. In Figure 5, we present the performance of SPACE trained ***over two iterations***. For example, the reported average performance 51.48 is measure on the model trained with a total of 150K samples, including 50K annotated data and 100K synthetic data. To ensure a fair comparison under the same computational budget, we run SFT for multiple rounds with annotated samples, and present the results in the following table. The results show that, even when using more annotated data, SFT still underperforms compared to SPACE (51.48 with a total of 150K samples).
>
> ||400K|600K|800K|
> | -----| -------| -------| -------|
> |SFT|50.48|50.14|49.84|
>
> ---
>
> **Q4**: Missing recent methods
>
> **A4**: Thanks for the helpful reminder. DNPO and SPACE share a similar motivation, both of which aim to address the model update stagnation issue in SPIN, but they adopt different strategies. DNPO proposes to selectively filter generated samples for continual updates, whereas SPACE introduces a novel optimization objective that ensures provably stable convergence (shown in ***Theorem 3***). We will include a more detailed discussion in the revised version.
>
> ---
>
> **Q5**: What happens if synthetic data quality degrades?
>
> **A5**: Thanks for the insightful question. The experiments in Figure 4 demonstrate that using synthetic samples generated by a weaker model leads to limited performance improvement, even with more training epochs. For this reason, SPACE employs an iterative strategy that updates synthetic samples using the most recent model for progressive evolution. We will include a more detailed discussion in the revised version.
>
> ---
>
> **Q6**: Could the iterative process gradually degrade capabilities in non-targeted areas?
>
> **A6**: Thanks for the valuable question. We acknowledge that this concern is well-founded, since self-play methods are designed to handle the scenarios with scarce annotated data, where fine-tuning LLMs while maintaining diversity over multiple benchmarks is inherently challenging. In our experiments, we observe that SPACE indeed demonstrates relatively marginal performance improvements on certain benchmarks, such as Winogrande. Notably, this is ***a common issue*** for self-play methods, not just specific to our SPACE. One possible explanation is that self-play methods gradually reinforce their attention to annotated data during iterations, which thereby reduces the diversity of outputs. Nevertheless, we would like to emphasize that SPACE is proposed to address the instability issue of SPIN, and we consider the investigation of diversity for self-play methods a meaningful direction and will explore it in the future.
>
> ---
>
> **Referece.**
>
> [1] Generative adversarial nets. 2014.
>
> [2] Mastering the game of go without human knowledge. 2017.
>
> [3] Dynamic Noise Preference Optimization for LLM Self-Improvement via Synthetic Data. 2025.
>
> [4] Measuring mathematical problem solving with the math dataset. 2021.
>
> [5] Can a suit of armor conduct electricity? a new dataset for open book question answering. 2018.
>
> [6] Judging llm-as-a-judge with mt-bench and chatbot arena. 2023.
>
> ---
>
> We hope that our responses can address your concerns, and we are also happy to provide further clarifications during the reviewer-author discussions if needed.

---

### Official Review · Reviewer_BDge · 2025-07-03

**Clarity:** 3
**Significance:** 3
**Originality:** 3
**Rating:** 5
**Confidence:** 3

**Summary:**

This paper introduces SPACE, a novel self-play fine-tuning method for large language models (LLMs) that leverages noise contrastive estimation (NCE) to address the instability issue in existing gap-based self-play approaches. SPACE treats synthetic samples as auxiliary components and discriminates them from real data via binary classification, independently optimizing absolute reward values for both types of data. Theoretically, the authors prove that SPACE converges stably to the real-world data distribution. Empirically, SPACE outperforms SPIN and SFT with more data, on diverse tasks from the HuggingFace Open LLM Leaderboard. It demonstrates stable performance evolution across iterations, avoiding the degradation seen in gap-based methods, and highlights the effectiveness of NCE in stabilizing self-play fine-tuning for LLMs with limited annotated data.

**Questions:**

See weaknesses above

**Ethical Concerns:**

["NO or VERY MINOR ethics concerns only"]

**Final Justification:**

My concerns have been addressed

**Limitations:**

yes

**Quality:**

3

**Strengths And Weaknesses:**

**Strengths:**
1. The paper points out an interesting reason why SPIN stops improving quickly, gives a reasonable explanation, and then comes up with a new loss function based on NCE. The new loss function has a good theoretical basis.
2. The results on various benchmarks show that it's significantly better than SPIN and its variants, and it seems to keep getting better over more rounds. So the method is good both in theory and in practice.
3. I like the analysis experiments, they're complete. They rule out possible counfounding factors that could explain why SPACE works better, like comparing more training rounds on the data from iteration 0. Also, the sample efficiency shown in Figure 5 is impressive, and I like how the results are presented.
4. The analysis of the reward decline problem is interesting too. It shows that SPACE properly reduces the problem, which helps explain why SPACE works well.

**Weaknesses:**
1. When looking at the gradient in Eq. 9, we can see it's similar to the policy gradient in RL. Since SPACE uses annotated data, it might also be related to off-policy RL. Given how much attention RL is getting lately, it would have been good to discuss this connection.
2. Why does SPACE do so much better than SPIN on GSM8K and IFEval? With such big improvements, some explanation or examples would be good.
3. The method only looks at one model, so it's not clear if it works for other models.
4. There's no proof for the theorems. They should at least be in the appendix.

---

> ### Author Rebuttal · Authors · 2025-07-30
>
> **Many thanks for the constructive reviews!**
>
> ---
>
> **Q1**: Connection to off-policy RL.
>
> **A1**: Thanks for the valuable suggestion! In the standard policy gradient formulation (e.g., REINFORCE [1]), the gradient is formulated as $\nabla_\theta J(\theta) = \mathbb{E}\_{u\sim p_\theta}[R_u \cdot \nabla_\theta \ln p_{\theta}(u|x)]$, where $R_u$ denotes the reward for a response $u$. Intuitively, (9) shares a similar structure to this formulation, if we set $R_u=\sigma_{\mu^{-1}}(-r(x,u))(p_{data}(u|x) - p_{\theta_t}(u|x))$. However, we would like to emphasize that SPACE is based on the self-play fine-tuning framework, where the goal is to iteratively improve model performance through two-player competition. This fundamentally differs from the exploration-exploitation trade-off in RL. Additionally, SPACE optimizes (8) in a supervised-learning manner, without involving explicit rewarding or rollout phrases, which are essential components of standard RL methods. We will include a more detailed discussion in the revised version.
>
>
> ---
>
> **Q2**: Why does SPACE do so much better than SPIN on GSM8K and IFEval?
>
> **A2**: Thanks for the insightful question. GSM8K and IFEval evaluates the abilities of mathematical problem solving and instruction following, respectively. Both benchmarks require strong logical reasoning and accurate responses. Based on the experimental results, we consider that for self-play fine-tuning methods, stable evolution during iterations is the key to ensuring rigorous reasoning and precise responses. Moreover, we present response comparisons on GSM8K below. From the responses, we observe that Mistral exhibits strong fluency but lacks understanding of the problem. SPIN demonstrates partial understanding but often fails to solve the problem correctly, likely due to unstable optimization. In contrast, SPACE, with its stable iterative evolution, is able to effectively solve practical problems.
>
> > **Prompt**: Weng earns \$12 an hour for babysitting. Yesterday, she just did 50 minutes of babysitting. How much did she earn?
> >
> > **Mistral**:  Guidance\\n\\nIn the last concept, we learned how to solve proportions involving rates. In this concept, we will be solving proportions as well. The difference is that rather than solving for a rate which compares two rates, we will be solving for an unknown quantity in a proportion. The formula to find any unknown amount is to cross multiply.\\n\\n#### Example A\\n\\nIf 24 is to 88 as 5 is to 12, how much is the unknown value 5?\\n\\nSolution: To find the value of an unknown amount in a proportion, we cross multiply the amount given in the proportion and the amount we are solving for... \\n\\nYou must check all answers, so check the answer of 12. ... The answer of 12 checks.\\n\\n#### Example B\\n\\n ... #### Concept Problem Revisited\\n\\nWeng earns 12 an hour for babysitting. Yesterday, she just did 50 minutes of babysitting. How much did she earn?\\n\\nThis is a proportional situation.\\n
> >
> > **SPIN**: If the word problem asks you to find the total number of dollars earned (the solution), then first, you need to translate the words into a math problem. When you read the problem, ask yourself, “What is the unknown?” In this case, it is the number of dollars earned.\\n\\nThe money earning rate is given.\\n\\nWeng earns 12 an hour for babysitting.\\n\\nThe time taken is known.\\n\\nYesterday, she just did 50 minutes of babysitting.\\n\\nWhat is the multiplication formula for calculating the total sum from rate and time?\\n\\nrate × time \= total\\n\\n12 × \\underline{\\underline{\\phantom{\\rule{1em}{0ex}}}} \= total\\n\\nWrite the question word to fill the missing part of the equation.\\n\\nMultiply the rate and time.\\n\\nHere, the unknown is total, which is also unknown to find.\\n\\nYou can find the answer to the problem by using the multiplication formula and the information given in the word problem.\\n\\nNote: 1 hour \= 60 minutes\\n\\nRate × time \= total\\n\\n12\\⨯50\=1,200\\n\\nShe earned 1,200.
> >
> > **SPACE**:  ## The math Rate \= (\\$12) per hour Number of hours \= (50 \\div  60) \= (0.833333) h Earned \= Rate x Number of hours \= (\\$12 \times 0.833333)  \= (\\$10) Therefore, Weng earned (\$10) for babysitting.
>
> ---
>
> **Q3**: The method only looks at one model, so it's not clear if it works for other models.
>
> **A3**: Thanks for the helpful suggestion. In fact, we have included the experiments on Zephyr in the ***Supplementary Material***, and the results also demonstrate the superiority of SPACE over SPIN. Additionally, we also conduct empirical studies on Qwen2.5-7B-base model. The results are presented in the table below, where *base model* refers to the average performance of Qwen2.5 over multiple tasks (the same task set as shown in Table 5; only average scores are shown due to character limitations). From the results, we observe that (i) SPIN continues to suffer from unstable iterative optimization, while SPACE demonstrates consistent performance improvements; and (ii) the performance gains achieved by SPACE on Qwen2.5 are less pronounced compared to those on Mistral and Zephyr. This may be because Qwen2.5 has already been well optimized during pretraining, leaving less room for further improvement when only 50K annotated examples are used.
>
> ||base model|iter0|iter1|iter2|iter3|iter4|
> | -------| ------------| -------| -------| -------| -------| -------|
> |SPIN|59.18|59.61|59.81|59.63|59.69|59.89|
> |SPACE|59.18|60.06|60.46|60.53|60.54|60.54|
>
> ---
>
> **Q4**: There's no proof for the theorems.
>
> **A4**: Thanks for the kind reminder. The proofs of all theorems are included in the ***Supplementary Material***, and we will highlight this in the revised version.
>
> ---
>
> **Reference.**
>
> [1] Simple statistical gradient following algorithms for connectionist reinforcement learning. 1992.
>
>
> ---
>
> We hope that our responses can address your concerns, and we are also happy to provide further clarifications during the reviewer-author discussions if needed.

---

> > ### Comment · Reviewer_BDge · 2025-08-06
> >
> > Thank you for the detailed response. I think this well-suited paper for NeurIPS and I've raised my score.

---

> > > ### Author Response · Authors · 2025-08-06
> > >
> > > Many thanks for your appreciation and support!  We will revise our paper according to your valuable suggestions.

---

### Official Review · Reviewer_1euE · 2025-07-03

**Clarity:** 3
**Significance:** 3
**Originality:** 4
**Rating:** 5
**Confidence:** 4

**Summary:**

This paper proposes SPACE, a self-play fine-tuning framework for large language models that avoids the instability of prior reward-gap–based methods like SPIN. SPACE reformulates the objective using a noise contrastive estimation (NCE) loss that computes a symmetric log-probability ratio between the current model and a frozen past model. This avoids the reward-gap collapse observed in SPIN as models converge. The authors provide theoretical guarantees that SPACE retains non-vanishing gradients and is preference-optimal under decomposable reward assumptions. Empirically, SPACE outperforms SPIN, DPO, and supervised fine-tuning on GSM8K and IFEval, offering smoother training dynamics and achieving alignment without human-labeled preferences.

**Questions:**

# Questions

1. How does SPACE affect response diversity and exploration?
    Does it suppress generation diversity by continually pushing away from previous outputs? Could this lead to degenerate or overly cautious outputs over time?

**Ethical Concerns:**

["NO or VERY MINOR ethics concerns only"]

**Final Justification:**

Based on a careful reading of the paper and the rebuttal, I believe this is a solid work suitable for publication at NeurIPS. My questions were satisfactorily addressed, and I am maintaining my original score.

**Limitations:**

Yes

**Quality:**

4

**Strengths And Weaknesses:**

# Strengths

1. The paper replaces reward-gap comparisons with a log-probability contrastive formulation, avoiding instability due to shrinking reward signals.
2. Theoretical Soundness: Proves consistency with preference optimization and non-vanishing gradients, even as the model improves over iterations.
3. Empirical Performance: Demonstrates strong results on reasoning tasks like GSM8K and IFEval using self-play without human preference data, outperforming SPIN and achieving performance comparable to or better than DPO in several settings.
4.Training Stability: SPACE maintains more stable and consistent performance gains across training iterations, whereas SPIN suffers from reward-gap collapse—an issue that limits the scalability of self-play fine-tuning.

# Weaknesses

1. Assumes Frozen Baseline Quality: Performance depends on the frozen past model being appropriately calibrated. If too weak or too strong, the contrastive learning signal may be ineffective.
2.No Diversity Evaluation: The paper does not analyze whether SPACE encourages mode collapse or overly conservative outputs, particularly as self-play proceeds across iterations. Output diversity is a key quality consideration in alignment methods and could affect robustness
3. Limited Benchmark Coverage: Focused on factual, single-turn tasks (GSM8K, IFEval); the behavior of SPACE in open-ended settings like dialogue or summarization is unexplored.

---

> ### Author Rebuttal · Authors · 2025-07-30
>
> **Many thanks for the constructive reviews!**
>
> ---
>
> **Q1**: Assumes Frozen Baseline Quality.
>
> **A1**: Thanks for the insightful comment. The quality of the reference model is indeed a crucial factor in self-play fine-tuning for LLM, and SPACE is designed with this in mind. In particular, we consider two scenarios: (i) where the reference model is weak, and (ii) where it is strong. For the first case, Figure 4 shows that using synthetic samples generated by a weaker model leads to suboptimal performance, even with more training epochs. For this reason, SPACE employs an iterative strategy that updates synthetic samples using the most recent model for progressive evolution. In the second case, a strong reference model may lead to unstable optimization (as observed in SPIN), and our SPACE is specifically designed to deal with this issue. Moreover, Theorem 3 shows that SPACE can preserve optimality even with strong reference model. We will include a detailed discussion in the revised version.
>
> ---
>
> **Q2**: Diversity and exploration of SPACE.
>
> **A2**: We appreciate the thoughtful question. We acknowledge that this concern is well-founded, since self-play methods are designed to handle the scenarios with scarce annotated data, where fine-tuning LLMs while maintaining diversity over multiple benchmarks is inherently challenging. In our experiments, we observe that SPACE indeed demonstrates relatively marginal performance improvements on certain benchmarks, such as Winogrande. Notably, this is ***a common issue*** for self-play methods, not just specific to our SPACE. One possible explanation is that self-play methods gradually reinforce their attention to annotated data during iterations, which thereby reduces the diversity of outputs. Nevertheless, we would like to emphasize that SPACE is proposed to address the instability issue of SPIN, and we consider the investigation of diversity for self-play methods a meaningful direction and will explore it in the future.
>
> ---
>
> **Q3**: The behavior of SPACE in open-ended settings like dialogue or summarization is unexplored.
>
> **A3**: Thanks for the helpful suggestion! We have conducted additional experiments on MATH [1], openbookqa [2], and MT-Bench [3], which evaluate mathematical reasoning, common knowledge, and conversational ability in open-ended scenarios, respectively. We present the results in the following table, where columns 3 to 7 correspond to iterations 0 to 4 of SPACE. From the table, we observe that SPACE can progressively improve performance while maintaining stability.
>
> ||Mistral|iter0|iter1|iter2|iter3|iter4|
> | ------------| ---------| -------| -------| -------| -------| -------|
> |MATH|3.25|3.40|3.58|3.69|3.84|3.90|
> |openbookqa|44.0|44.4|45.0|45.4|45.4|46.0|
> |MT-Bench|3.56|5.87|6.13|6.27|6.43|6.44|
>
> ---
>
> **Reference.**
>
> [1] Measuring mathematical problem solving with the math dataset. 2021.
>
> [2] Can a suit of armor conduct electricity? a new dataset for open book question answering. 2018.
>
> [3] Judging llm-as-a-judge with mt-bench and chatbot arena. 2023.
>
> ---
>
> We hope that our responses can address your concerns, and we are also happy to provide further clarifications during the reviewer-author discussions if needed.

---

> > ### Comment · Reviewer_1euE · 2025-08-06
> >
> > Thank you for the response and the extra experiments (suggesting to be included in the supplementary). I think this paper is a good fit for NeurIPS. I stick to my previous rating.

---

> > > ### Author Response · Authors · 2025-08-07
> > >
> > > We sincerely appreciate your recognition and support! We will incorporate your constructive suggestions into the revised version.

---

### Note · Authors · 2025-08-12

Many thanks for constructive comments from all reviewers! We appreciate the opportunity to briefly summarize the reviewer-author discussion.

---
**Our work**. This paper proposes a novel self-play method SPACE to address the instability issue of existing methods. The key idea is to treat synthetic samples as auxiliary components and discriminate them from real ones in a binary classification manner. As a result, SPACE independently optimizes the absolute reward values for each type of data, ensuring stable evolution. Theoretically, SPACE enjoys provable achievability (**Theorem 2**) and maintainability (**Theorem 3**) for the optimal distribution. Empirically, SPACE significantly improves performances of various LLMs (**Tables 1 and 4**) and outperforms SFT that uses more annotated data (**Figure 5**).

**Reviews and rebuttal**. We are grateful for reviewers' valuable comments and particularly their recognition of our work, including theoretical soundness (Reviewers 1euE, BDge), promising performances (Reviewers 1euE, BDge, Hm2z, j1nk) and complete analysis (Reviewer BDge). In rebuttal, we provide detailed responses to each concern, especially performance on other tasks, advanced LLMs, and comparisons with SFT. Below, we briefly summarize these clarifications:
- **Performance on other tasks**. We validate performance on additional tasks (MATH, openbookqa and MT-Bench). The results show that SPACE still enjoys progressively improved performance with stable evolution (**A3 to Reviewer 1euE,**  **A2 to Reviewer Hm2z,**   **A6 to Reviewer j1nk**).
- **Performance on advanced LLMs**. We conduct additional experiments on Qwen2.5-7B, and results show significant superiority of SPACE over SPIN. We also analyze potential reasons for relatively smaller improvements of self-play methods on Qwen2.5 compared to other LLMs: (i) limited room for further improvement; (ii) scarce annotated training data (**A3 to Reviewer BDge,**  **A3 to Reviewer j1nk**).
- **Comparisons with SFT**. As suggested by reviewers, we provide a fairer comparison between SPACE and SFT with multiple epochs, and results show that SPACE still achieves better performance (**A3 to Reviewer Hm2z,**  **A4 to Reviewer j1nk**).

We are pleased that most reviewers confirmed that our rebuttal has addressed their concerns during discussions, and we will incorporate their valuable suggestions in the revised version. Once again, we sincerely appreciate the efforts of reviewers and ACs in evaluating our work.

---

### Decision · Program_Chairs · 2025-09-17

**Decision:**

Accept (poster)

**Comment:**

This paper presents a novel self-play fine-tuning method inspired by noise contrastive estimation. The paper identifies a failure mode of a previous approach, SPIN, and shows that a noise contrastive estimation approach would mitigate the problem. The proposed method is sound and principled, and the paper is written clearly. The main concerns raised during the discussion include 1) only having a single base model in their experiment, 2) limited benchmark coverage, and 3) fairer comparison with SFT. Also, other issues, including diversity and connection to other prior works, are discussed. These concerns were addressed in the authors' response. Overall, the paper appears to be a solid contribution to the community, and I recommend acceptance. The authors are strongly encouraged to incorporate the feedback from the reviewers in the updated manuscript.